# Flow-Attentional Graph Neural Networks

**Pascal Plettenberg**  *plettenberg@uni-kassel.de*
*Intelligent Embedded Systems, University of Kassel*

**Dominik Köhler**  *dkoehler@uni-kassel.de*
*Intelligent Embedded Systems, University of Kassel*

**Bernhard Sick**  *bsick@uni-kassel.de*
*Intelligent Embedded Systems, University of Kassel*

**Josephine M. Thomas**  *thomasj@uni-greifswald.de*
*GAIN Group, Institute of Data Science, University of Greifswald*

**Reviewed on OpenReview:** *https://openreview.net/forum?id=tOzg7UxTPD*

## Abstract

Graph Neural Networks (GNNs) have become essential for learning from graph-structured data. However, existing GNNs do not consider the conservation law inherent in graphs associated with a flow of physical resources, such as electrical current in power grids or traffic in transportation networks, which can lead to reduced model performance. To address this, we propose *flow attention*, which adapts existing graph attention mechanisms to satisfy Kirchhoff's first law. Furthermore, we discuss how this modification influences the expressivity and identify sets of non-isomorphic graphs that can be discriminated by flow attention but not by standard attention. Through extensive experiments on two flow graph datasets—electronic circuits and power grids—we demonstrate that flow attention enhances the performance of attention-based GNNs on both graph-level classification and regression tasks.

## 1 Introduction

Graph neural networks (GNNs) (Scarselli et al., 2008; Kipf & Welling, 2017) have emerged as a powerful framework that extends the scope of deep learning from Euclidean to graph-structured data, which is prevalent across many real-world domains, such as social networks (Fan et al., 2019), recommender systems (Wu et al., 2022), materials science (Reiser et al., 2022), or epidemiology (Liu et al., 2024). Especially attention-based GNNs have become increasingly popular due to their ability to select relevant features adaptively (Sun et al., 2023). As graph data becomes increasingly common, advancing GNN architectures is crucial for improving performance in tasks such as node classification (Hamilton et al., 2017), graph regression (Gilmer et al., 2017), or link prediction (Zhang & Chen, 2018).

In many important applications of GNNs, graphs are naturally associated with a flow of physical resources, such as electrical current in electronic circuits (Sánchez et al., 2023) or power grids (Liao et al., 2021), traffic in transportation networks (Jiang & Luo, 2022), water in river networks (Sun et al., 2021), or raw materials and goods in supply chains (Kosasih & Brintrup, 2022). All nodes in these *resource flow graphs*, except for source and target nodes, are subject to Kirchhoff's first law, which states that the sum of all incoming and outgoing flows must be zero, reflecting the conservation of resources. By contrast, *informational graphs*—such as computation graphs, social networks, or citation networks—are not associated with any physical flow but rather represent relationships or information transfer. Information can be arbitrarily duplicated in these graphs, unlike in flow graphs, where such duplication would violate the conservation law.

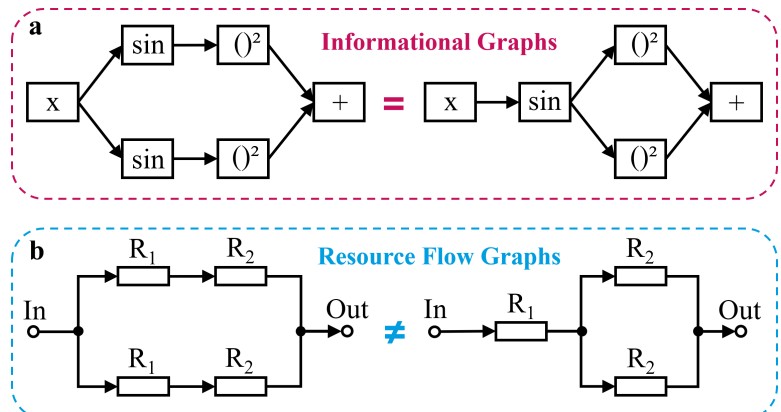

Figure 1: Two non-isomorphic graphs that are equivalent as informational graphs, but different as resource flow graphs. **a** The two different directed graph structures represent the same computation (example adapted from Zhang et al. (2019)). **b** The same graph structures as above represent different electronic circuits.

As a result, two non-isomorphic graphs may be equivalent as informational graphs but non-equivalent as flow graphs. For example, in a computational graph, the output of a sine operation can be freely duplicated. Thus, the two non-isomorphic graph structures in Fig. 1a represent the same computation. However, the same graph structures may also represent electronic circuits governed by Kirchhoff's first law (see Fig. 1b). In this case, the two circuits are different because combining or splitting resistors changes their electrical properties.

Electronic circuits and other resource flow graphs can often be represented as directed acyclic graphs (DAGs). However, GNNs specifically tailored to DAGs typically encode the rooted trees of the output nodes rather than the exact graph structure. Hence, if two DAGs exhibit the same computation tree (e.g., the graphs in Fig. 1), they cannot be distinguished, which limits the performance in many graph learning tasks. While these graph structures might be interchangeable for pure "information" tasks, a sufficiently expressive GNN should map them to distinct representations when performing tasks where physical flows are relevant.

**Main Contributions.** Inspired by the conservation law in resource flow graphs, we propose *flow attention* on graphs, which normalizes attention scores across outgoing neighbors instead of incoming ones. This simple but effective modification can be applied to any attention-based GNN and allows the model to better capture the physical flow of a graph. We discuss the expressivity of the resulting models and demonstrate that flow attention enables the discrimination of any DAG from its computation tree. Based on this observation, we propose *FlowDAGNN*, a flow-attentional GNN for DAGs. Finally, we conduct extensive experiments on multiple datasets, including cascading failure analysis on power grids and property prediction on electronic circuits, covering undirected graphs and DAGs. Our results indicate that flow attention improves the performance of attention-based GNNs on graph-level classification and regression tasks. [1]

## 2 Related Work

In recent years, many new GNN models have been specifically designed for different graph types (Thomas et al., 2023). Despite their fundamental differences, informational graphs and flow graphs are mainly treated by the same basic message-passing layers, such as GraphSAGE (Hamilton et al., 2017), GAT (Veličković et al., 2018), or GIN (Xu et al., 2019). In these models, messages exchanged between neighboring nodes do not depend on the number of recipients. Instead, the information is arbitrarily duplicated and passed to all neighbors. GCN (Kipf & Welling, 2017) applies a symmetric neighborhood normalization but exhibits limited expressivity and performance. Attention-based GNNs, such as GAT, GATv2 (Brody et al., 2022), or UniMP (Shi et al., 2021), adaptively weight neighboring node features, leading to improved representation

---

[1]The code is available at `https://github.com/pasplett/FlowGNN`.

learning. However, the attention weights are obtained through normalization across incoming neighbors, allowing for arbitrary message duplication. Our approach normalizes across outgoing neighbors instead, which better captures the physical flow of a graph while preserving the benefits of graph attention.

Many flow graphs, including the example graphs in Fig. 1, can be naturally expressed as DAGs, e.g., operational amplifiers (Dong et al., 2023) or material flow networks (Perera et al., 2018). Before GNNs were introduced, recursive neural networks were applied to DAGs (Sperduti & Starita, 1997; Frasconi et al., 1998), and contextual recursive cascade correlation was proposed to overcome limitations in expressivity (Hammer et al., 2005). However, these early works lacked the advantages of modern GNNs, which have been extended to DAGs in recent years (Zhang et al., 2019; Thost & Chen, 2021). In directed acyclic GNNs, nodes are typically updated sequentially following the partial order of the DAG, and the final target node representations are used as the graph embedding. Although these sequential models outperform undirected GNNs on DAG datasets, they still aggregate node neighborhood information similarly. This means they only encode the computation tree of the output nodes and not the exact structure of the DAG, resulting in limited expressivity.

A possible approach to overcome the problem of indistinguishable flow graphs is to use node indices or random features as input node features (Loukas, 2020; Sato et al., 2021), which enables the model to uniquely identify each node. However, the resulting GNN model is no longer permutation invariant, which reduces its generalization capability. Similar problems arise for Transformer-based models (Vaswani et al., 2017) such as PACE (Dong et al., 2022), which incorporate the relational inductive bias (Battaglia et al., 2018) via positional encodings. A different strategy would be to introduce Kirchhoff's first law through an additional physics-informed loss term (Donon et al., 2020), which considerably increases the training complexity and is only useful if the target variable is the resource flow itself. Our approach enhances the expressivity of attention-based GNNs on flow graphs while preserving permutation invariance and computational efficiency.

## 3 Preliminaries

**Graph.** A directed graph can be defined as a tuple $\mathcal{G} = (\mathcal{V}, \mathcal{E})$ containing a set of nodes $\mathcal{V} \subset \mathbb{N}$ and a set of directed edges $\mathcal{E} \subseteq \mathcal{V} \times \mathcal{V}$. Thereby, we define $e = (u, v)$ as the directed edge from node $u$ to node $v$. An edge is called undirected if $(u, v) \in \mathcal{E}$ whenever $(v, u) \in \mathcal{E}$. Furthermore, we call the set $\mathcal{N}_{\text{in}}(v) = \{u \in \mathcal{V} \mid (u, v) \in \mathcal{E}\}$ the incoming neighborhood of $v$ and the set $\mathcal{N}_{\text{out}}(v) = \{u \in \mathcal{V} \mid (v, u) \in \mathcal{E}\}$ the outgoing neighborhood of $v$.

**Node Multiset.** For each node in a graph, the feature vectors of a set of incoming nodes can be represented as a multiset (Xu et al., 2019). A multiset is a pair $(S, m)$, where $S$ is a set of distinct elements (the node features) and $m : S \to \mathbb{N}$ is the multiplicity of each element. We call two multisets $X_1 = (S, m_1)$, $X_2 = (S, m_2)$ equally distributed if $m_2 = k \cdot m_1$ with $k \in \mathbb{N}_{\geq 1}$.

**Directed Acyclic Graph.** A directed graph without cycles is called a directed acyclic graph (DAG). In the context of DAGs, we call the incoming neighborhood the predecessors of a node, and the outgoing neighborhood the successors of a node. The set of all ancestors of node $v$ contains all nodes $u \in \mathcal{V}$ such that $v$ is reachable from $u$. Similarly, the descendants are the nodes $u \in \mathcal{V}$ that are reachable from $v$. Finally, the set of nodes without predecessors is called the set of start or initial nodes, denoted by $\mathcal{I} \subset \mathcal{V}$, and the set of nodes without successors is called the set of end or final nodes, denoted by $\mathcal{F} \subset \mathcal{V}$.

**Computation Tree.** Let $\mathcal{D} = (\mathcal{V}, \mathcal{E}, r)$ be a rooted DAG with a unique final node called the root $r$. Its computation tree $\gamma(\mathcal{D})$ is obtained by leaving each node $v$ with at most one successor, and for any node $v$ with $n \geq 2$ successors, replacing $v$ by $n$ copies $v_1, \ldots, v_n$, each connected to exactly one of $v$'s successors and inheriting all of $v$'s incoming edges. This procedure yields a rooted tree with the same root $r$. See App. D for a visualization.

**Flow Graph.** Let $\mathcal{G} = (\mathcal{V}, \mathcal{E})$ be a graph and $\mathcal{S}, \mathcal{T} \subseteq \mathcal{V}$ be the sources and targets of $\mathcal{G}$. A flow on $\mathcal{G}$ is a mapping $\psi : \mathcal{E} \to \mathbb{R}$ that satisfies Kirchhoff's first law:

$$\sum_{u \in \mathcal{N}_{\text{in}}(v)} \psi(u, v) = \sum_{u \in \mathcal{N}_{\text{out}}(v)} \psi(v, u) \quad \forall\, v \in \mathcal{V} \setminus \{\mathcal{S}, \mathcal{T}\}. \tag{1}$$

If a graph is associated with a flow $\psi$ as defined above, we refer to it as a flow graph. In DAGs, the start nodes are sources, and the end nodes are targets: $\mathcal{I} \subseteq \mathcal{S}$ and $\mathcal{F} \subseteq \mathcal{T}$.

**Graph Neural Networks.** Graph Neural Networks (GNNs) transfer the concept of traditional neural networks to graph data. Thereby, the node representations $\{\boldsymbol{h}_i \in \mathbb{R}^\rho \mid i \in \mathcal{V}\}$ with the feature dimension $\rho$ are updated iteratively by aggregating information from neighboring nodes via message-passing. The updated node representations $\boldsymbol{h}'_i$, i.e., the output of the network layer, are given by

$$\boldsymbol{h}'_i = \phi \left( \boldsymbol{h}_i, \bigoplus_{j \in \mathcal{N}_{\text{in}}(i)} f\left(\boldsymbol{h}_j\right) \right), \tag{2}$$

with a learnable message function $f$, an aggregator $\oplus$, e.g., sum or mean, and an update function $\phi$. The choice of $\phi$, $\oplus$, and $f$ defines the design of a specific GNN model.

**Directed Acyclic Graph Neural Networks.** The main idea of GNNs for DAGs is to process and update the nodes sequentially according to the partial order defined by the DAG. Thereby, the update of a node representation $\boldsymbol{h}_i$ is computed based on the current-layer node representations of node $i$'s predecessors. The message-passing scheme of directed acyclic GNNs can therefore be expressed as

$$\boldsymbol{h}'_i = \phi \left( \boldsymbol{h}_i, \bigoplus_{j \in \mathcal{N}_{\text{in}}(i)} f\left(\boldsymbol{h}'_j\right) \right). \tag{3}$$

The most widely used directed acyclic GNNs are D-VAE (Zhang et al., 2019) and DAGNN (Thost & Chen, 2021), the latter of which uses standard attention for aggregation. Both models utilize gated recurrent units (GRU) as the update function $\phi$ and are briefly explained in App. B. As an alternative to sequential models, DAGs can be encoded using Transformer-based architectures, such as PACE (Dong et al., 2022) or DAGformer (Luo et al., 2023).

## 4    Graph Attention and its Limitations

In this section, we demonstrate why standard graph attention is insufficient for flow graphs. First, we show that standard attention cannot distinguish node neighborhoods with equal distribution of node features, which generally limits its expressivity. Next, we prove that attention-based directed acyclic GNNs cannot discriminate between a DAG and its computation tree. As a result, they cannot distinguish non-isomorphic DAGs that exhibit the same computation tree, e.g., the example graphs from Fig. 1.

### 4.1    Attentional GNNs

An attentional GNN uses a scoring function $e : \mathbb{R}^\rho \times \mathbb{R}^\rho \to \mathbb{R}$ to compute attention coefficients

$$e_{ij} = e\left(\boldsymbol{h}_i, \boldsymbol{h}_j\right), \tag{4}$$

indicating the importance of the features of node $j$ to node $i$. The computed attention coefficients $e_{ij}$ are normalized across all incoming neighboring nodes $j$ using softmax:

$$\alpha_{ij} = \text{softmax}_j(e_{ij}) = \frac{\exp(e_{ij})}{\sum_{k \in \mathcal{N}_{\text{in}}(i)} \exp(e_{ik})}. \tag{5}$$

Finally, the aggregation corresponds to a weighted average of the incoming messages:

$$\boldsymbol{h}'_{\text{att},i} = \phi \left( \sum_{j \in \mathcal{N}_{\text{in}}(i)} \alpha_{ij} f\left(\boldsymbol{h}_{\text{att},j}\right) \right). \tag{6}$$

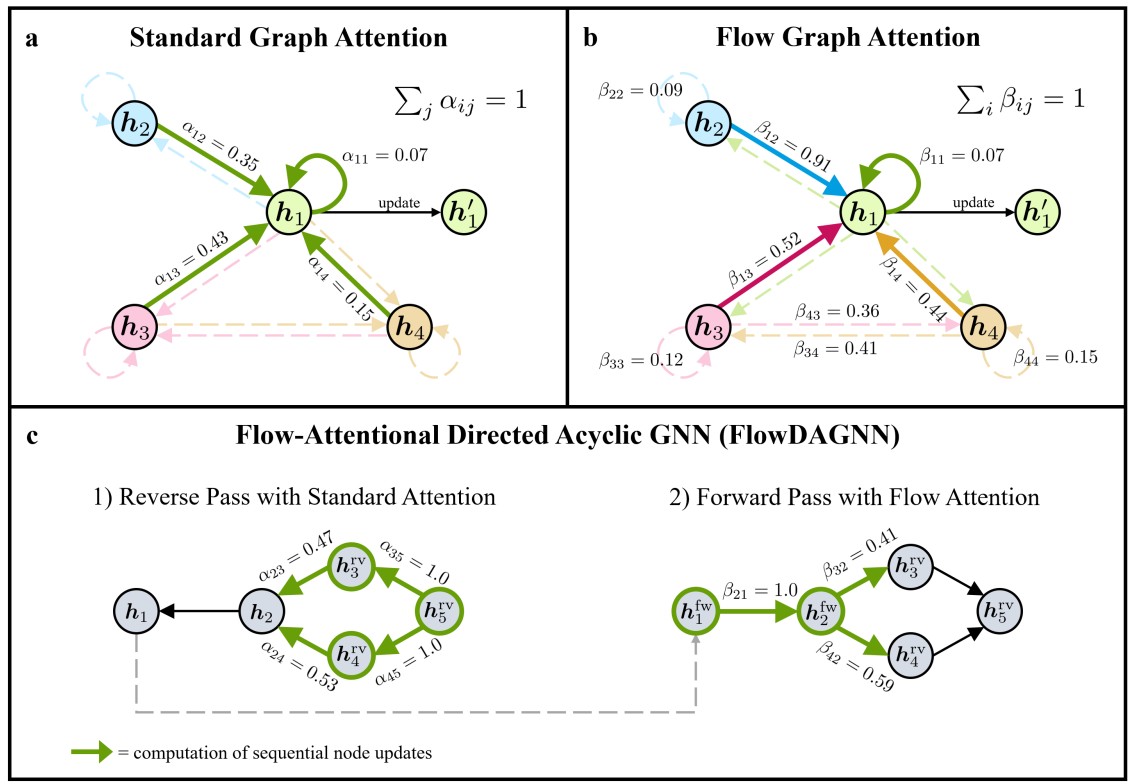

Figure 2: **a** Standard graph attention mechanism as it is applied in attentional GNNs. The attention weights associated with edges of the same color sum to 1. **b** The proposed flow attention mechanism applied in FlowGNNs. The flow attention weights associated with edges of the same color sum to 1. **c** Two snapshots during the reverse and forward pass of FlowDAGNN. Nodes marked in green have already been updated.

Popular attentional GNNs include GAT (Veličković et al., 2018), GATv2 (Brody et al., 2022), and TransformerConv (TC) [2] (Shi et al., 2021), which mainly differ in the choice of the scoring function $e$ (see App. C). The graph attention mechanism is visualized in Fig. 2a.

The weighted mean aggregation limits the expressivity of attention-based GNNs. Similar to the mean aggregator, standard attention does not capture the exact node neighborhood but the distribution of nodes in the neighborhood.

**Lemma 4.1.** *Assume $\mathcal{X}_1 = (S, m)$ and $\mathcal{X}_2 = (S, k \cdot m)$ are multisets with the same distribution, with $k \in \mathbb{N}_{\geq 1}$. Then $\boldsymbol{h}'_{att}(\mathcal{X}_1) = \boldsymbol{h}'_{att}(\mathcal{X}_2)$, for any choice of $\phi$ and $f$.*

Proofs of all Lemmas and Corollaries can be found in the Appendix A.

## 4.2 Attention on DAGs

The node update of an attentional directed acyclic GNN can be expressed as

$$\boldsymbol{h}'_{\text{att},i} = \phi \left( \boldsymbol{h}_{\text{att},i}, \sum_{j \in \mathcal{N}_{\text{in}}(i)} \alpha_{ij} f \left( \boldsymbol{h}'_{\text{att},j} \right) \right). \tag{7}$$

A directed acyclic GNN sequentially updates each node's feature vector until it arrives at one or more final nodes. If there are multiple final nodes, their representations can be gathered into a single virtual node.

---

[2]Please note that TransformerConv is not a Graph Transformer with a global attention module (such as GraphGPS (Rampášek et al., 2022)) but a message-passing GNN with a local attention mechanism adopted from the vanilla Transformer (Vaswani et al., 2017). It was originally introduced under the name *Graph Transformer* as part of the UniMP model (Shi et al., 2021).

Since all other nodes are ancestors of this final node, its representation can be used to characterize the whole DAG. The computation history of the final node representation can be visualized as a rooted subtree, which means that if two non-isomorphic DAGs represent the same computation, the rooted subtree structures of their final node representations are equivalent. Therefore, since standard attention weights only depend on incoming neighborhoods, directed acyclic GNNs with standard attention mechanisms cannot discriminate a DAG $\mathcal{D}$ from its computation tree $\gamma(\mathcal{D})$ (see Fig. 3 in App. D).

**Corollary 4.2.** *For every DAG $\mathcal{D} \in \Delta$ and every directed acyclic GNN $\mathcal{A}_{att}$ with a standard attention mechanism, it holds:*

$$\mathcal{A}_{att}(\mathcal{D}) = \mathcal{A}_{att}(\gamma(\mathcal{D})).$$

As a consequence, an attentional directed acyclic GNN cannot distinguish between the two graphs in Fig. 1, since they exhibit the same computation tree. Note that Corollary 4.2 is also valid for directed acyclic GNNs with other aggregators that are independent of outgoing node neighborhoods, e.g., sum or mean.

## 5 The Flow Attention Mechanism

In this section, we introduce the flow attention mechanism and discuss its influence on expressivity. Next, we define flow attention on DAGs, which enables the discrimination of a DAG from its computation tree. Finally, we propose FlowDAGNN, a directed acyclic GNN model for flow graphs.

### 5.1 Flow-Attentional GNNs

Standard graph attention mechanisms normalize attention scores across all incoming edges. Therefore, a message does not depend on the number of nodes it is forwarded to and can be duplicated arbitrarily, contradicting the resource conservation concept inherent in flow graphs. Therefore, we propose an alternative graph attention mechanism that normalizes the attention scores across outgoing edges instead (see Fig. 2b). We denote the resulting *flow attention weights* as $\beta_{ij}$ to distinguish them from the standard attention weights $\alpha_{ij}$:

$$\beta_{ij} = \text{softmax}_i(e_{ij}) = \frac{\exp(e_{ij})}{\sum_{k \in \mathcal{N}_{\text{out}}(j)} \exp(e_{kj})}. \tag{8}$$

Although the attention scores are normalized across outgoing edges, we still aggregate incoming messages in order to update the hidden state of node $i$:

$$\boldsymbol{h}'_{\text{flow},i} = \phi \left( \sum_{j \in \mathcal{N}_{\text{in}}(i)} \beta_{ij} f\left(\boldsymbol{h}_{\text{flow},j}\right) \right). \tag{9}$$

However, since the messages are multiplied with the flow attention weights $\beta_{ij}$, they now also depend on the neighborhood of the message's sender, i.e., node $j$. In this way, we ensure that a message transmitted by any node cannot be duplicated arbitrarily but instead is distributed among all outgoing neighbors. We define a flow-attentional graph neural network (FlowGNN) as a modified version of an attentional GNN, which utilizes the flow attention mechanism from Eq. 9 for aggregating node neighborhood information instead of standard attention. Furthermore, we denote the corresponding FlowGNN versions of standard attentional GNNs as FlowGAT, FlowGATv2, FlowTC, etc.

The flow attention weights determine how a node's message is distributed among its outgoing neighbors, i.e., $\beta_{ij}$ can be seen as the relative flow from node $j$ to node $i$. An absolute flow $\psi_\beta$ can be calculated by iteratively multiplying subsequent flow attention weights in a graph.

**Lemma 5.1.** *Let $\mathcal{G} = (\mathcal{V}, \mathcal{E})$ be a graph with fixed source nodes $\mathcal{S} \subset \mathcal{V}$ and target nodes $\mathcal{T} \subset \mathcal{V}$. We define $\psi_\beta$ for every directed edge $(j, i) \in \mathcal{E}$ as*

$$\psi_\beta(j, i) = \begin{cases} \beta_{ij} \sum_{k \in \mathcal{N}_{in}(j)} \psi_\beta(k, j), & \text{if } j \in \mathcal{V} \setminus \{\mathcal{S}, \mathcal{T}\} \\ \psi_0(j, i), & \text{otherwise.} \end{cases}$$

*Thereby, $\psi_0(j,i)$ is the absolute outgoing flow from a source or target node. Then $\psi_\beta$ is a flow on $\mathcal{G}$ that satisfies Kirchhoff's first law.*

Since the absolute flow $\psi_\beta$ satisfies Kirchhoff's first law, a FlowGNN is capable of implicitly taking into account the underlying resource flow of a graph via the flow attention weights $\beta_{ij}$. Another advantage of the flow attention mechanism is that the aggregation over incoming neighbors is not a weighted mean but a weighted sum. Therefore, in contrast to standard attention, flow attention can discriminate between multisets with the same distribution.

**Lemma 5.2.** *Assume $\mathcal{X}_1 = (S, m)$ and $\mathcal{X}_2 = (S, k \cdot m)$ are multisets with the same distribution of elements for some $k \in \mathbb{N}_{\geq 2}$. Furthermore, assume that all nodes $s \in S$ have the same outgoing neighborhood $\mathcal{N}_{out}(s)$ in $\mathcal{X}_1$ and $\mathcal{X}_2$. Then $\exists \phi, f$ such that $\boldsymbol{h}'_{flow}(\mathcal{X}_1) \neq \boldsymbol{h}'_{flow}(\mathcal{X}_2)$.*

Therefore, flow-attentional GNNs are particularly well-suited for tasks where not only statistical information but also the precise graph structure plays a crucial role. This is especially true when features occur repeatedly, e.g., in the case of multiple identical resistors in an electronic circuit.

### 5.2   Flow-Attention on DAGs

We define the node update for a flow-attentional directed acyclic GNN similar to Eq. 7:

$$\boldsymbol{h}'_{\text{flow},i} = \phi\left(\boldsymbol{h}_{\text{flow},i}, \sum_{j \in \mathcal{N}_{\text{in}}(i)} \beta_{ij} f\left(\boldsymbol{h}'_{\text{flow},j}\right)\right). \tag{10}$$

If all nodes in a graph have only one outgoing edge, e.g., the graph is a rooted tree, then $\beta_{ij} = 1 \, \forall \, i, \, j$. In this case, the flow attention mechanism degenerates to a sum aggregation, which can be maximally expressive for the right choice of $\phi$ and $f$ (Xu et al., 2019).

**Corollary 5.3.** *Let $T \subset \Delta$ be the subset of all DAGs, where each node has at most one outgoing edge (i.e., the set of all rooted trees). Then, for any two rooted trees $\mathcal{T}_1$, $\mathcal{T}_2$ with $\mathcal{T}_1 \neq \mathcal{T}_2$, there exists a flow-attentional directed acyclic GNN $\mathcal{A}_{flow}$ such that:*

$$\mathcal{A}_{flow}(\mathcal{T}_1) \neq \mathcal{A}_{flow}(\mathcal{T}_2).$$

Furthermore, contrary to standard attention, flow-attentional directed acyclic GNNs can distinguish between graphs and their message-passing computation trees.

**Theorem 5.4.** *For every DAG $\mathcal{D} \in \Delta$ with $\mathcal{D} \notin \mathcal{T}$ (i.e., a true DAG) and its computation tree $\gamma(\mathcal{D})$, there exists a flow-attentional directed acyclic GNN $\mathcal{A}_{flow}$ such that:*

$$\mathcal{A}_{flow}(\mathcal{D}) \neq \mathcal{A}_{flow}(\gamma(\mathcal{D})).$$

*Proof.* We take an arbitrary but fixed intermediate node $i$ of the DAG and denote its representation under the flow-attentional directed acyclic GNN by

$$\boldsymbol{h}_i = \boldsymbol{h}^{\mathcal{D}}_{\text{flow},i} \quad \text{and} \quad \tilde{\boldsymbol{h}}_i = \boldsymbol{h}^{\gamma(\mathcal{D})}_{\text{flow},i},$$

for the graph $\mathcal{D}$ and its computation tree $\gamma(\mathcal{D})$, respectively. Since $\mathcal{D}$ is not a tree, there is at least one node $j$ with more than one successor. Hence, there is a predecessor $j$ for which $\beta_{ij} < 1$. We then need to show that $\boldsymbol{h}'_i \neq \tilde{\boldsymbol{h}}'_i$:

$$\phi\Big(\boldsymbol{h}_i, \sum_{j \in \mathcal{N}_{\text{in},i}} \beta_{ij} f\big(\boldsymbol{h}'_j\big)\Big) \neq \phi\Big(\tilde{\boldsymbol{h}}_i, \sum_{j \in \mathcal{N}_{\text{in},i}} f\big(\tilde{\boldsymbol{h}}'_j\big)\Big).$$

By choosing $f$ in the following way:

$$\exists f : f(\tilde{\boldsymbol{h}}'_j) \geq f(\boldsymbol{h}'_k) \quad \forall j, k \in \mathcal{N}_{\text{in},i}, \tag{11}$$

it follows due to $\exists j : \beta_{ij} < 1$:

$$\sum_{j \in \mathcal{N}_{\mathrm{in},i}} \beta_{ij} f(\boldsymbol{h}'_j) < \sum_{j \in \mathcal{N}_{\mathrm{in},i}} f(\boldsymbol{h}'_j) \overset{\text{Eq. 11}}{\leq} \sum_{j \in \mathcal{N}_{\mathrm{in},i}} f(\tilde{\boldsymbol{h}}'_j).$$

Hence, with choosing $\phi$ to be injective, $\mathcal{A}_{\mathrm{flow}}$ distinguishes $\mathcal{D}$ from $\gamma(\mathcal{D})$. $\qquad\square$

From Theorem 5.4, we conclude that a flow-attentional directed acyclic GNN can discriminate the example DAGs from Fig. 1 for the right choice of $\phi$ and $f$. In practice, we can model the composition $f^{(l)} \circ \phi^{(l-1)}$ on the $l$-th GNN layer by a universal approximator, e.g., GRU (Schäfer & Zimmermann, 2006; hoon Song et al., 2023).

## 5.3 FlowDAGNN

We propose a flow-attentional version of the attention-based DAGNN (Thost & Chen, 2021). A naive approach would be to simply replace the attention weights $\alpha_{ij}$ with flow attention weights $\beta_{ij}$. Due to the sequential nature of directed acyclic GNNs, the computation of the flow attention weight $\beta_{ij}$ in a DAG only depends on the node $i$ and all its ancestors. However, in contrast to standard attention weights, flow attention weights are forward-directed, which means that they should also be conditioned on all descendants of the node $i$. Analogously, the electrical current splitting up from one node into multiple branches depends on the whole branch and not only on the first node in each branch. In App. E, we give a simple example of an electronic circuit illustrating this situation.

Hence, we construct a FlowDAGNN layer from two sublayers (see Fig. 2c). In the first sublayer (we call it the *reverse pass*), we invert all edges of the DAG $\mathcal{D}$ and apply a standard DAGNN layer to the reverse DAG $\tilde{\mathcal{D}}$. This is equivalent to performing the aggregation over all successor nodes in the original DAG $\mathcal{D}$ instead of over all predecessors:

$$\boldsymbol{m}_i^{\mathrm{rv}} = \sum_{j \in \mathcal{N}_{out}(i)} \alpha_{ij} \left( \boldsymbol{h}_i, \boldsymbol{h}_j^{\mathrm{rv}} \right) \boldsymbol{h}_j^{\mathrm{rv}}, \tag{12}$$

$$\alpha_{ij} \left( \boldsymbol{h}_i, \boldsymbol{h}_j^{\mathrm{rv}} \right) = \operatorname*{softmax}_{j \in \mathcal{N}_{out}(i)} \left( (\boldsymbol{w}_1^{\mathrm{rv}})^{\mathrm{T}} \boldsymbol{h}_i + (\boldsymbol{w}_2^{\mathrm{rv}})^{\mathrm{T}} \boldsymbol{h}_j^{\mathrm{rv}} \right), \tag{13}$$

$$\boldsymbol{h}'_i = \boldsymbol{h}_i^{\mathrm{rv}} = \mathrm{GRU}(\boldsymbol{h}_i, \boldsymbol{m}_i^{\mathrm{rv}}). \tag{14}$$

In the second sublayer, we perform a *forward pass* on the original DAG $\mathcal{G}$. However, this time we are applying the flow attention mechanism described in Section 5.1 to compute flow attention weights $\beta_{ij}$:

$$\boldsymbol{m}_i^{\mathrm{fw}} = \sum_{j \in \mathcal{N}_{out}(i)} \beta_{ij} \left( \boldsymbol{h}_i^{\mathrm{rv}}, \boldsymbol{h}_j^{\mathrm{fw}} \right) \boldsymbol{h}_j^{\mathrm{fw}}, \tag{15}$$

$$\beta_{ij} \left( \boldsymbol{h}_i^{\mathrm{rv}}, \boldsymbol{h}_j^{\mathrm{fw}} \right) = \operatorname*{softmax}_{i \in \mathcal{N}_{out}(j)} \left( (\boldsymbol{w}_1^{\mathrm{fw}})^{\mathrm{T}} \boldsymbol{h}_i^{\mathrm{rv}} + (\boldsymbol{w}_2^{\mathrm{fw}})^{\mathrm{T}} \boldsymbol{h}_j^{\mathrm{fw}} \right), \tag{16}$$

$$\boldsymbol{h}_i^{\mathrm{fw}} = \mathrm{GRU}(\boldsymbol{h}_i^{\mathrm{rv}}, \boldsymbol{m}_i^{\mathrm{fw}}). \tag{17}$$

Since the hidden states $\boldsymbol{h}_i^{\mathrm{rv}}$ of the reverse pass contain information about all descendants of the node $i$, and the hidden states $\boldsymbol{h}_j^{\mathrm{fw}}$ contain information about all ancestors of the node $j$, the computation of the flow attention weights $\beta_{ij}$ essentially takes into account information about all nodes of the graph that are connected to the node $i$.

After $L$ FlowDAGNN layers, we compute the graph-level representation from both the reverse pass representations of the start nodes as well as the forward pass representations of the end nodes and concatenate across layers:

$$\boldsymbol{h}_{\mathcal{G}} = \operatorname*{Max-Pool}_{i \in \mathcal{I}} \left( \overset{L}{\underset{l=0}{\|}} \boldsymbol{h}_i^{\mathrm{rv},l} \right) \Big\| \operatorname*{Max-Pool}_{j \in \mathcal{F}} \left( \overset{L}{\underset{l=0}{\|}} \boldsymbol{h}_j^{\mathrm{fw},l} \right). \tag{18}$$

## 5.4 Applicability Beyond Flow Graphs

The flow attention mechanism can be integrated into any GNN with a local attention mechanism. Thus, flow-attentional GNNs can technically be applied to the same graph datasets as their base models, which means their applicability is not necessarily limited to flow graphs. However, it might be less effective on other types of graph datasets.

In flow attention, attention scores are normalized across outgoing neighbors, guiding a model to learn how to distribute a node's message among them. This is especially useful on flow graphs governed by Kirchhoff's first law, such as power grids, electronic circuits, or traffic networks. In contrast, other types of graph datasets, such as citation graphs or social networks, do not have an inherent structure that requires an information distribution across outgoing nodes. For example, there may be nodes that broadcast information to many adjacent nodes, e.g., influencers in social networks (Chen et al., 2009).

Furthermore, as pointed out in the Introduction, some non-isomorphic graph structures might be equivalent in the context of informational graphs (e.g., computational graphs) and therefore should be mapped to the same representation, a consistency not guaranteed by a flow-attentional GNN due to Theorem 5.4. We therefore expect FlowGNNs to be less effective on these types of graph datasets.

On the other hand, the flow-attention mechanism has the advantage over standard attention that it can distinguish between node multisets with the same distribution (Lemma 5.2). This could lead to performance improvements on graphs where certain node features occur repeatedly, which is often the case in flow graph applications (e.g., multiple identical components in electronic circuits), but can also be the case outside of the flow graph context (e.g., multiple identical atoms in molecules).

In this study, we focus on evaluating the effectiveness of the flow attention mechanism on flow graph datasets, including power grids and electronic circuits. However, additional experimental results on standard graph datasets can be found in App. G.3.

## 6 Experiments

We perform two different experiments. First, we perform graph-level multiclass classification on undirected flow graphs, comparing the effectiveness of our flow attention mechanism against standard graph attention. In the second experiment, we perform graph regression on DAGs to compare our proposed FlowDAGNN model with relevant directed acyclic GNN baselines.

### 6.1 Graph Classification on Undirected Flow Graphs

**Dataset.** We use the publicly available power grid data from the PowerGraph benchmark dataset (Varbella et al., 2024), which encompasses the IEEE24, IEEE39, IEEE118, and UK transmission systems. The graphs contained in these datasets are undirected and cyclic and represent test power systems mirroring real-world power grids. The test systems differ in scale and topology, covering various relevant parameters. Further details can be found in App. F.

**Task.** We perform cascading failure analysis as a graph-level multiclass classification task. Thereby, we utilize the attributed graphs provided by the PowerGraph dataset, each representing unique pre-outage operating conditions along with a set of outages corresponding to the removal of a single or multiple branches. An outage may result in demand not served (DNS) by the grid, and a cascading failure may occur, meaning that one or more additional branches trip after the initial outage. The model is supposed to predict whether the grid is stable (DNS = 0 MW) or unstable (DNS > 0 MW) after the outage, and additionally, whether a cascading failure occurs, resulting in four distinct categories representing the possible combinations of stable/unstable and cascading failure yes/no.

**Models and Baselines.** We take three widely used attention-based GNNs (GAT, GATv2, and Transformer-Conv (TC)) and compare them against the corresponding flow-attentional variants FlowGAT, FlowGATv2, and FlowTC. Additionally, we compare against three popular non-attentional GNN baselines (GCN, Graph-SAGE, and GIN). For each model, we perform a hyperparameter optimization by varying the number of message-passing layers (1, 2, 3) and the hidden dimension (8, 16, 32). Between subsequent message-passing layers, we apply the ReLU activation function followed by a dropout of 10%. To obtain graph embeddings

Table 1: **Test set balanced accuracy** ($\%$,$\uparrow$) and **F1-score (macro-averaged,** $\%$,$\uparrow$**)** for the cascading failure analysis (multiclass classification) on four different power grid test systems from the PowerGraph dataset. Reported results represent the average over five training runs with different random seeds, along with the standard deviation. The best result for each test system is marked in bold.

| | IEEE24 | | IEEE39 | | IEEE118 | | UK | |
|---|---|---|---|---|---|---|---|---|
| **MODEL** | Bal. Acc. ($\uparrow$) | Macro-F1 ($\uparrow$) | Bal. Acc. ($\uparrow$) | Macro-F1 ($\uparrow$) | Bal. Acc. ($\uparrow$) | Macro-F1 ($\uparrow$) | Bal. Acc. ($\uparrow$) | Macro-F1 ($\uparrow$) |
| GCN | $89.4 \pm 1.0$ | $89.7 \pm 1.0$ | $81.5 \pm 5.1$ | $81.9 \pm 5.1$ | $78.2 \pm 6.0$ | $77.7 \pm 6.7$ | $88.4 \pm 1.2$ | $86.3 \pm 1.6$ |
| GraphSAGE | $95.6 \pm 0.2$ | $95.4 \pm 0.4$ | $88.7 \pm 4.6$ | $89.1 \pm 4.5$ | $98.7 \pm 0.1$ | $98.4 \pm 0.1$ | $95.2 \pm 0.3$ | $95.2 \pm 0.3$ |
| GIN | $98.0 \pm 0.8$ | $97.3 \pm 0.9$ | $95.2 \pm 1.7$ | $94.6 \pm 1.4$ | $96.5 \pm 2.5$ | $96.3 \pm 2.5$ | $\mathbf{97.1 \pm 0.2}$ | $96.4 \pm 0.2$ |
| GAT | $90.8 \pm 4.1$ | $90.6 \pm 3.6$ | $90.9 \pm 2.7$ | $90.0 \pm 2.7$ | $92.1 \pm 1.5$ | $91.2 \pm 1.5$ | $94.9 \pm 1.0$ | $94.3 \pm 1.3$ |
| **FlowGAT** | $93.3 \pm 1.1$ | $93.0 \pm 1.0$ | $94.1 \pm 1.3$ | $93.2 \pm 1.3$ | $96.2 \pm 2.7$ | $96.1 \pm 2.6$ | $94.9 \pm 0.4$ | $94.3 \pm 0.4$ |
| GATv2 | $91.9 \pm 4.0$ | $90.0 \pm 4.7$ | $87.8 \pm 2.0$ | $86.3 \pm 2.2$ | $92.6 \pm 1.6$ | $91.7 \pm 1.3$ | $95.3 \pm 0.7$ | $94.8 \pm 1.0$ |
| **FlowGATv2** | $96.1 \pm 0.9$ | $95.6 \pm 0.8$ | $95.9 \pm 0.7$ | $\mathbf{95.0 \pm 0.9}$ | $\mathbf{99.1 \pm 0.1}$ | $\mathbf{99.0 \pm 0.1}$ | $96.9 \pm 0.3$ | $96.4 \pm 0.4$ |
| TC | $97.3 \pm 0.5$ | $96.7 \pm 0.6$ | $90.7 \pm 2.3$ | $90.8 \pm 2.0$ | $98.7 \pm 0.1$ | $98.4 \pm 0.1$ | $96.5 \pm 0.3$ | $96.2 \pm 0.2$ |
| **FlowTC** | $\mathbf{98.7 \pm 0.3}$ | $\mathbf{98.1 \pm 0.3}$ | $\mathbf{96.0 \pm 0.6}$ | $94.4 \pm 0.6$ | $98.8 \pm 0.2$ | $98.6 \pm 0.2$ | $\mathbf{97.1 \pm 0.4}$ | $\mathbf{96.8 \pm 0.5}$ |

from the node embeddings, we apply a global maximum pooling operator. For the final prediction, we use a single linear layer or a two-layer perceptron with a LeakyReLU activation, depending on which type of prediction layer was used for the corresponding model in the original PowerGraph benchmark.

The message-passing GNNs compared in this experiment are important components of many recent Graph Transformer models Shehzad et al. (2024), which combine local message-passing with global attention. In App. G.2, we report additional experimental results with recent state-of-the-art Graph Transformer models: SAT (Chen et al., 2022), GraphGPS (Rampášek et al., 2022), and Exphormer (Shirzad et al., 2023). We will investigate the integration of flow-attentional GNNs into Graph Transformer models in future work.

**Experimental Setting.** We stick closely to the original benchmark setting in Varbella et al. (2024) by splitting the datasets into train/validation/test with ratios 85/5/10% and using the Adam optimizer (Kingma, 2014) with an initial learning rate of $10^{-3}$ as well as a scheduler that reduces the learning rate by a factor of five if the validation accuracy plateaus for ten epochs. The negative log-likelihood is used as the loss function, and balanced accuracy (Brodersen et al., 2010) is used as the primary evaluation metric due to the strong class imbalance (see App. F). We train all models with a batch size of 16 for a maximum number of 500 epochs and apply early stopping with a patience of 20 epochs. Each training run is repeated five times with different random seeds.

**Results.** The balanced accuracies and macro-F1 scores on the test set are reported for each model on each of the four test systems in Tab. 1. We only report the results for the best model architecture from the hyperparameter optimization. Thereby, we noticed that the accuracy mostly improves with more message-passing layers, which has already been observed for power grid data in Ringsquandl et al. (2021). The FlowGNNs perform better than their corresponding standard GNN version in most cases. FlowGAT shows a higher balanced accuracy compared to GAT for the test systems IEEE24, IEEE39, and IEEE118, and a comparable performance on the UK test system. In the case of GATv2, the FlowGNN version even outperforms its standard counterpart on all test systems. FlowTC performs better than TC on all test systems except for IEEE118, where it shows a comparable performance. On all test systems, the best-performing model among the tested ones is a flow-attentional GNN. Overall, these results indicate that the flow attention mechanism, which is the only applied change to the corresponding baselines, can enhance the performance of attention-based GNNs on undirected flow graph data.

## 6.2 Graph Regression on Directed Acyclic Flow Graphs

**Dataset.** We utilize the Ckt-Bench101 dataset from the publicly available Open Circuit Benchmark (OCB) (Dong et al., 2023), which was developed to evaluate methods for electronic design automation. The dataset contains 10,000 operational amplifiers (Op-Amps) represented as DAGs and provides circuit specifications (e.g., gain and bandwidth) obtained from simulations. Details can be found in App. F.

Table 2: **Test set RMSE** and **Pearson's R** (%) for the prediction of three different Op-Amp properties from the Ckt-Bench101 dataset. Reported results represent the average over ten training runs with different random seeds, along with the standard deviation. The best result for each property is marked in bold.

| | GAIN | | BANDWIDTH | | FoM | |
|---|---|---|---|---|---|---|
| **MODEL** | RMSE ($\downarrow$) | Pearson's R ($\uparrow$) | RMSE ($\downarrow$) | Pearson's R ($\uparrow$) | RMSE ($\downarrow$) | Pearson's R ($\uparrow$) |
| PACE | $0.253 \pm 0.009$ | $97.1 \pm 0.3$ | $0.443 \pm 0.014$ | $90.9 \pm 0.5$ | $0.443 \pm 0.009$ | $90.8 \pm 0.5$ |
| DAGformer (SAT) | $0.234 \pm 0.012$ | $97.2 \pm 0.3$ | $0.459 \pm 0.010$ | $89.2 \pm 0.4$ | $0.450 \pm 0.015$ | $89.6 \pm 0.7$ |
| D-VAE | $0.229 \pm 0.004$ | $97.3 \pm 0.1$ | $0.430 \pm 0.008$ | $90.6 \pm 0.3$ | $0.421 \pm 0.011$ | $91.0 \pm 0.4$ |
| DAGNN | $0.215 \pm 0.002$ | $97.6 \pm 0.0$ | $0.396 \pm 0.008$ | $92.1 \pm 0.3$ | $0.396 \pm 0.005$ | $92.0 \pm 0.2$ |
| **FlowDAGNN** | $\mathbf{0.209 \pm 0.007}$ | $\mathbf{97.8 \pm 0.1}$ | $\mathbf{0.371 \pm 0.008}$ | $\mathbf{93.1 \pm 0.3}$ | $\mathbf{0.366 \pm 0.008}$ | $\mathbf{93.3 \pm 0.3}$ |

**Task.** We perform graph-level regression to predict the properties of the Op-Amps. For this purpose, we train three separate instances of each model for the prediction of gain, bandwidth, and figure of merit (FoM), respectively. The FoM is a measure of the circuit's overall performance and depends on gain, bandwidth, and phase margin.

**Models and Baselines.** We compare our proposed model, FlowDAGNN, against widely used baseline models from the literature, including GNN- and Transformer-based models tailored to DAGs: D-VAE, DAGNN, DAGformer (building upon the Structure-Aware Transformer (SAT, Chen et al. (2022))), and PACE. Thereby, we use the default parameters from Dong et al. (2023) where applicable, and from the original publications elsewhere, as well as the model-specific readout layers. For FlowDAGNN, we use two layers as described in Sec. 5.3 (each comprising one reverse and one forward pass) and adopt all other model parameters from DAGNN. The final prediction is done using a two-layer perceptron with a ReLU activation in between. Right before these final layers, we apply a dropout of 50% for regularization.

**Experimental setting.** We split the dataset into train/validation/test with ratios 80/10/10% and select the same test set as in Dong et al. (2023). Furthermore, we use the AdamW optimizer (Loshchilov, 2017) with an initial learning rate of $10^{-4}$ and train each model using the mean squared error (MSE) as the loss function with a batch size of 64 for a maximum of 500 epochs, but apply early stopping with a patience of 20 epochs. Each training run is repeated ten times with different random seeds.

**Results.** The RMSEs on the test set for all models and all OpAmp target properties are presented in Tab. 2. Among all tested models, FlowDAGNN shows the best performance on all target properties. Especially, it performs better than DAGNN, the standard attentional model on which it is originally based. Thereby, FlowDAGNN performs slightly better in predicting the gain property and significantly better in the other two properties.

# 7 Conclusion

In this paper, we proposed the flow attention mechanism, which adapts existing standard graph attention mechanisms to be more suitable for learning tasks on flow graph datasets, where the flow of physical resources (e.g., electricity) plays an important role. Inspired by Kirchhoff's first law, the mechanism normalizes attention scores across outgoing edges instead of incoming ones, which avoids unrestricted message duplication and better captures the underlying physical flow of the graph. We discussed the influence of this architectural change on the model expressivity and showed that flow-attentional GNNs, in contrast to GNNs using standard attention, can distinguish node neighborhoods with the same distribution. Since the proposed modification of the standard graph attention is simple and minimal, it can be easily implemented in practice and does not significantly increase the computational effort (see App. G.4 for more details on computational efficiency).

Since many flow graphs can be naturally expressed as directed acyclic graphs (DAGs), we also extended the flow attention mechanism to DAGs and proposed a specific model, namely FlowDAGNN. We proved that this model can distinguish non-isomorphic directed acyclic graphs that were so far indistinguishable for existing GNNs tailored to DAGs. We validated our theoretical findings with extensive experiments on power grids

and electronic circuit datasets, including undirected graphs and DAGs, respectively. Our results indicate that the flow attention mechanism considerably improves the performance of their standard counterparts on graph-level regression and classification tasks.

In the future, we want to analyze how the proposed models scale to larger circuits and power grids. Another interesting direction will be to evaluate the performance on node- and edge-level tasks, as well as on other flow graph data, such as traffic networks or supply chains. Finally, we also want to investigate the usage of flow-attentional GNNs as local message-passing components in Graph Transformer models (see also App. G.2).

### Acknowledgments

This work was funded by the German Federal Ministry of Research, Technology and Space (16ME0877). We also acknowledge the helpful feedback from Mohamed Hassouna, Clara Holzhüter, Lukas Rauch, and Jan Schneegans.

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

## A Proofs

*Proof of Lemma 4.1.* We are given the node update of an attentional GNN from Equation 6, adapted to the multiset $\mathcal{X} = (S, m)$ as input:

$$\boldsymbol{h}'_{\text{att}} = \phi \left( \sum_{s \in S} \alpha_{1s} m(s) f\left(\boldsymbol{h}_{\text{att},s}\right) \right),$$

with $\alpha_{1s}$ being the softmax over the edge importance scores:

$$\alpha_{1s} = \frac{\exp(e_{1s})}{\sum_{s' \in S} \exp(e_{1s'})}.$$

For the two given multisets $\mathcal{X}_1, \mathcal{X}_2$, $\boldsymbol{h}'_{\text{att},i}$ gets the same result for any choice of $\phi$ and $f$:

$$\begin{aligned}
\boldsymbol{h}'_{\text{att}}(\mathcal{X}_1) &= \phi \left( \frac{\sum_{s \in S} m(s) \exp(e_{1s}) f(\boldsymbol{h}_{\text{att},s})}{\sum_{s' \in S} m(s') \exp(e_{1s'})} \right) \\
&= \phi \left( \frac{\sum_{s \in S} k \cdot m(s) s \exp(e_{1s}) f(\boldsymbol{h}_{\text{att},s})}{\sum_{s' \in S} k \cdot m(s') \exp(e_{1s'})} \right) \\
&= \boldsymbol{h}'_{\text{att}}(\mathcal{X}_2).
\end{aligned}$$

$\square$

*Proof of Corollary 4.2.* This follows from the definition of a computation tree: Each node in $\gamma(\mathcal{D})$ gets the same representation as the corresponding node in $\mathcal{D}$, as the aggregation is carried out over the same multiset of node features and does not take into account the outgoing neighborhoods of the nodes contained in the multiset. Hence, the representation of the (virtual) output node is the same in both cases, leading to equal DAG representations. $\square$

*Proof of Lemma 5.1.* We recall the definition of $\psi_\beta$ from Lemma 5.1:

$$\psi_\beta(j, i) = \begin{cases} \beta_{ij} \displaystyle\sum_{k \in \mathcal{N}_{in}(j)} \psi_\beta(k, j), & \text{if } j \in \mathcal{V} \setminus \{\mathcal{S}, \mathcal{T}\} \\ \psi_0(j, i), & \text{otherwise.} \end{cases} \tag{19}$$

Since the flow attention weights correspond to the attention scores normalized across outgoing edges, it holds that

$$\sum_{i \in \mathcal{N}_{out}(j)} \beta_{ij} = 1.$$

Multiplying on both sides with the sum of $\Psi_\beta$ across all incoming edges of node $j$ gives:

$$\sum_{i \in \mathcal{N}_{out}(j)} \beta_{ij} \left( \sum_{k \in \mathcal{N}_{in}(j)} \psi_\beta(k, j) \right) = \sum_{k \in \mathcal{N}_{in}(j)} \psi_\beta(k, j).$$

Using the definition of $\psi_\beta$ from Eq. 19, we arrive at Kirchhoff's first law:

$$\sum_{i \in \mathcal{N}_{out}(j)} \psi_\beta(j, i) = \sum_{k \in \mathcal{N}_{in}(j)} \psi_\beta(k, j) \quad \forall j \in \mathcal{V} \setminus \{\mathcal{S}, \mathcal{T}\}.$$

$\square$

*Proof of Lemma 5.2.* We are given the node update of a flow-attentional GNN from Equation 9, adapted to the multiset $\mathcal{X} = (S, m)$ as input:

$$\boldsymbol{h}'_{\text{flow}} = \phi\left(\sum_{s \in S} \beta_{1s} m(s) f\left(\boldsymbol{h}_{\text{flow},s}\right)\right).$$

Since the elements of $S$ have the same features and outgoing neighborhoods in $\mathcal{X}_1$ and $\mathcal{X}_2$, the flow attention weights are the same in both cases. Therefore, if $\phi$ is injective, $\boldsymbol{h}'_{\text{flow},i}$ gets different results:

$$\boldsymbol{h}'_{\text{flow}}(\mathcal{X}_1) = \phi\left(\sum_{s \in X_1} \beta_{1s} m(s) f(\boldsymbol{h}_{\text{flow},s})\right)$$
$$\overset{k \geq 2, f \neq 0}{\neq} \phi\left(\sum_{s \in X_1} k\beta_{1s} m(s) f(\boldsymbol{h}_{\text{flow},s})\right)$$
$$= \boldsymbol{h}'_{\text{flow}}(\mathcal{X}_2).$$

$\square$

*Proof of Corollary 5.3.* We are given the node update of a flow-attentional GNN from Equation 9, adapted to the multisets $\mathcal{X}_i = (S_i, m_i), i \in \{1, 2\}$ as input:

$$\boldsymbol{h}'_{\text{att}} = \phi\left(\sum_{s \in S_i} \beta_{1s} m_i(s) f\left(\boldsymbol{h}_{\text{att},s_i}\right)\right),$$

with $\beta_{1s}$ being equal to 1, as all nodes have at most one outgoing edge in a rooted tree. Then, we fulfill the prerequisites of Lemma 5 from Xu et al. (2019), which states that for the right choice of $f$ and $\phi$, any two different multisets can be distinguished. Thus, $A_{\text{flow}}$ can distinguish the trees $T_1$ and $T_2$. $\square$

## B   Directed Acyclic GNN Baselines

In the encoder of the D-VAE model (Zhang et al., 2019), the aggregation corresponds to a gated sum using a mapping network $m$ and a gating network $g$, and the update function $\phi$ is a gated recurrent unit (GRU) (Cho et al., 2014):

$$\boldsymbol{m}'_i = \sum_{j \in \mathcal{N}_{in}(i)} g(\boldsymbol{h}'_j) \odot m(\boldsymbol{h}'_j), \tag{20}$$

$$\boldsymbol{h}'_i = \text{GRU}(\boldsymbol{h}_i, \boldsymbol{m}'_i). \tag{21}$$

Another popular model is the DAGNN (Thost & Chen, 2021), which also uses a GRU for the update function but the message function is an attention mechanism with model parameters $\boldsymbol{w}_1$ and $\boldsymbol{w}_2$:

$$\boldsymbol{m}'_i = \sum_{j \in \mathcal{N}_{in}(i)} \alpha_{ij}\left(\boldsymbol{h}_i, \boldsymbol{h}'_j\right) \boldsymbol{h}'_j, \tag{22}$$

$$\alpha_{ij} = \underset{j \in \mathcal{N}_{in}(i)}{\text{softmax}}\left(\boldsymbol{w}_1^{\text{T}} \boldsymbol{h}_i + \boldsymbol{w}_2^{\text{T}} \boldsymbol{h}'_j\right). \tag{23}$$

Since the embeddings of the (possibly multiple) end nodes contain information on the whole DAG, they are typically used for computing the graph-level representations. After $L$ layers, the graph-level embedding can be obtained by concatenating the end node representations from all layers followed by a max-pooling across all end nodes:

$$\boldsymbol{h}_{\mathcal{G}} = \underset{i \in \mathcal{F}}{\text{Max-Pool}}\left(\overset{L}{\underset{l=0}{\|}} \boldsymbol{h}_i^l\right). \tag{24}$$

As DAGs are treated as sequences by the above models, they can also be processed in reversed order by inverting the edges. Therefore, directed acyclic GNNs are also capable of bidirectional processing. Using the tilde notation to denote node representations in the reverse DAG, the readout function for bidirectional processing can then be expressed as:

$$\boldsymbol{h}_{\mathcal{G}} = \text{FC} \left( \underset{i \in \mathcal{I}}{\text{Max-Pool}} ( \overset{L}{\underset{l=0}{\Big\|}} \tilde{\boldsymbol{h}}_i^l) \;\Big\|\; \underset{j \in \mathcal{F}}{\text{Max-Pool}} ( \overset{L}{\underset{l=0}{\Big\|}} \boldsymbol{h}_j^l) \right). \tag{25}$$

Note that the representations of the forward and reverse processing are computed independently, which is different from the reverse and forward pass in FlowDAGNN. In all our experiments, we use bidirectional processing for D-VAE and DAGNN.

## C  Scoring Functions of Attentional GNN Baselines

In GAT (Veličković et al., 2018), the scoring function is defined as

$$e_{\text{GAT}} (\boldsymbol{h}_i, \boldsymbol{h}_j) = \text{LeakyReLU} \left( \boldsymbol{a}^T \cdot [\boldsymbol{W}\boldsymbol{h}_i \,\|\, \boldsymbol{W}\boldsymbol{h}_j] \right). \tag{26}$$

Thereby, the linear layers $\boldsymbol{a}$ and $\boldsymbol{W}$ are applied consecutively, making it possible to collapse them into a single linear layer.

In GATv2 (Brody et al., 2022), a strictly more expressive attention mechanism is proposed, in which the second linear layer $\boldsymbol{a}$ is applied *after* the nonlinearity:

$$e_{\text{GATv2}} (\boldsymbol{h}_i, \boldsymbol{h}_j) = \boldsymbol{a}^T \text{LeakyReLU} \left( \boldsymbol{W} \cdot [\boldsymbol{h}_i \,\|\, \boldsymbol{h}_j] \right). \tag{27}$$

Thus, GATv2 is effectively using a multi-layer perceptron (MLP) to compute the attention scores, allowing for dynamic attention compared to the static attention performed by GAT.

Finally, TransformerConv (TC) (Shi et al., 2021) is transferring the attention mechanism of the Transformer model (Vaswani et al., 2017) to graph learning:

$$\boldsymbol{q}_i = \boldsymbol{W}_q \boldsymbol{h}_i + \boldsymbol{b}_q, \tag{28}$$

$$\boldsymbol{k}_j = \boldsymbol{W}_k \boldsymbol{h}_j + \boldsymbol{b}_k, \tag{29}$$

$$e_{\text{TC}} (\boldsymbol{h}_i, \boldsymbol{h}_j) = \frac{\boldsymbol{q}_i^T \cdot \boldsymbol{k}_j}{\sqrt{d}}, \tag{30}$$

where $\boldsymbol{q}_i \in \mathbb{R}^d$ is the query vector, $\boldsymbol{k}_j \in \mathbb{R}^d$ is the key vector and $\boldsymbol{W}_q, \boldsymbol{W}_k, \boldsymbol{b}_q, \boldsymbol{b}_k$ are trainable parameters.

All of the above scoring functions can be extended to multi-head attention and can incorporate edge features as well. Furthermore, it is possible to include self-loops. These characteristics are naturally inherited by the corresponding FlowGNNs.

## D  DAGs and Computation Trees

Fig. 3 again shows the two different DAG structures from the examples in Fig. 1 together with their corresponding computation trees as defined in Section 3. Although the two DAGs are different, they have the same computation tree. Since standard attention weights are computed by normalizing over incoming neighbors, they are the same for both DAGs. However, the flow attention weights are obtained by normalizing across outgoing neighbors. Since the grey and blue nodes (colors represent distinct node features) exhibit different outgoing neighborhoods in the two DAGs, the flow attention weights are different. Thus, a flow-attentional directed acyclic GNN can distinguish the two DAGs while a standard attentional one cannot.

## E  Example Circuit Motivating the Reverse Pass in FlowDAGNN

Fig. 4 shows an example for an electronic circuit modeled as a DAG, which motivates the necessity for the reverse pass in FlowDAGNN. If FlowDAGNN would only compute node representations via the forward

| | DAG | Computation Tree | |
|---|---|---|---|
| | | with Attention Weights | with **Flow** Attention Weights |

Figure 3: Two non-isomorphic DAGs together with their corresponding computation trees, which are equivalent. Distinct node features are visualized by different colors. The middle and right columns show some example standard and flow attention weights. While the standard attention weights are always the same for both DAGs, the flow attention weights are different.

pass, the flow attention weight $\beta_{ij}$ would only depend on all ancestors of node $i$. This means that the edge from $In$ to $R_1$ in the upper branch would receive the same flow attention weight as the edge from $In$ to $R_1$ in the lower branch. However, since $R_1 \ll R_2$, the electrical current in the upper branch would be much smaller than the current in the lower branch. Therefore, FlowDAGNN would not be capable of modeling the electrical current flow via the flow attention weights. Only if the reverse pass is applied before the forward pass, the flow attention weights can also be conditioned on the descendants of node $i$.

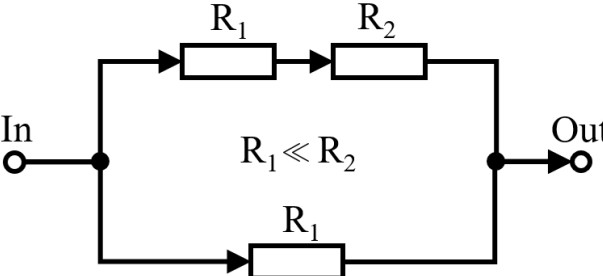

Figure 4: A simple example circuit described as a DAG, which explains why the reverse pass is necessary in FlowDAGNN. Without the reverse pass, the flow attention weights from the input node to the $R_1$ nodes would be identical, whereas the electrical current flow is different due to $R_1 \ll R_2$.

# F   Details on PowerGraph and Ckt-Bench101

**PowerGraph.** The PowerGraph dataset (Varbella et al., 2024) contains four different transmission systems (IEEE24, IEEE39, IEEE118, UK) with unique graph structures. For the cascading failure analysis, each test system was simulated for different operating conditions together with a specific initial outage, resulting in a

Table 3: Number of nodes and edges for each test system as well as the number of corresponding graph samples contained in the PowerGraph dataset (see Varbella et al. (2024)).

| Test system | No. Nodes | No. Edges | No. Graphs |
|---|---|---|---|
| IEEE24 | 24 | 38 | 21500 |
| IEEE39 | 39 | 46 | 28000 |
| IEEE118 | 118 | 186 | 122500 |
| UK | 29 | 99 | 64000 |

Table 4: Distribution of the classification labels for each test system in the PowerGraph dataset (see Varbella et al. (2024)). DNS stands for "demand not served" and c.f. stands for "cascading failure", corresponding to at least one more tripping branch after the initial outage.

| Test system | **Category A** DNS > 0 MW c. f. | **Category B** DNS > 0 MW no c. f. | **Category C** DNS = 0 MW c. f. | **Category D** DNS = 0 MW no c. f. |
|---|---|---|---|---|
| IEEE24 | 15.8% | 4.3% | 0.1% | 79.7% |
| IEEE39 | 0.55% | 8.4% | 0.45% | 90.6% |
| IEEE118 | >0.1% | 5.0% | 0.9% | 93.9% |
| UK | 3.5% | 0% | 3.8% | 92.7% |

large number of graph samples. The number of nodes and edges in each test system as well as the number of graph samples are reported in Tab. 3.

Tab. 4 shows how the classification labels are distributed in the PowerGraph dataset for each test system. Due to the strong class imbalance, the balanced accuracy BA is used as the evaluation metric (Brodersen et al., 2010), which is defined as the mean of sensitivity and specificity:

$$\text{BA} = \frac{1}{2}\left(\frac{\text{TP}}{\text{TP} + \text{FN}} + \frac{\text{TN}}{\text{TN} + \text{FP}}\right). \tag{31}$$

Here, TP/FP/TN/FN represent true/false positive/negative predictions.

**CktBench-101.** The CktBench-101 dataset from the Open Circuit Benchmark Dong et al. (2023) contains 10,000 artificially generated operational amplifiers represented as DAGs. Fig. 5 shows the distribution of the number of nodes and the number of edges among all graphs in the dataset. The average number of nodes is 9.6 with a standard deviation of 2.1. The average number of edges is 14.5 with a standard deviation of 5.3. We are using the most recent update of the CktBench-101 dataset, which does not contain any failed simulations anymore.

## G   Additional Experiments

### G.1   Correlation between (Flow) Attention Weights

Fig. 6 shows the Pearson correlation between the (flow) attention weights of different models trained on the four test systems contained in the PowerGraph dataset. Models with the same type of attention (e.g., GAT and GATv2, or FlowGAT and FlowTC) exhibit small to medium positive correlations (0.17 - 0.56) between their attention weights. On the other hand, while models with different types of attention (e.g., GAT and FlowGAT, or GATv2 and FlowTC) do not show any considerable correlations on the IEEE24 and UK test systems, small negative correlations can be observed between attention and flow attention weights on the IEEE39 and IEEE118 test systems. These small negative correlations can be explained by a higher number of links between nodes with very different node degrees, resulting in small standard attention weights and large flow attention weights or vice versa, depending on the edge direction. Apart from that, the difference

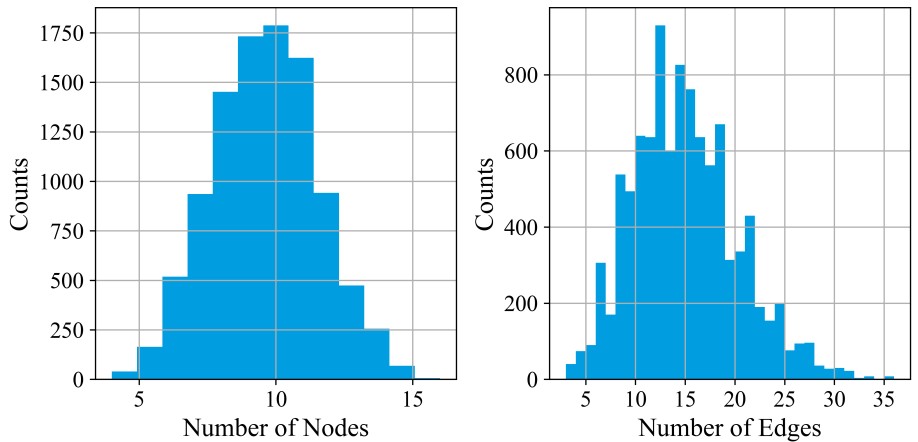

Figure 5: Distribution of the number of nodes (left) and number of edges (right) within the Ckt-Bench101 dataset (Dong et al., 2023).

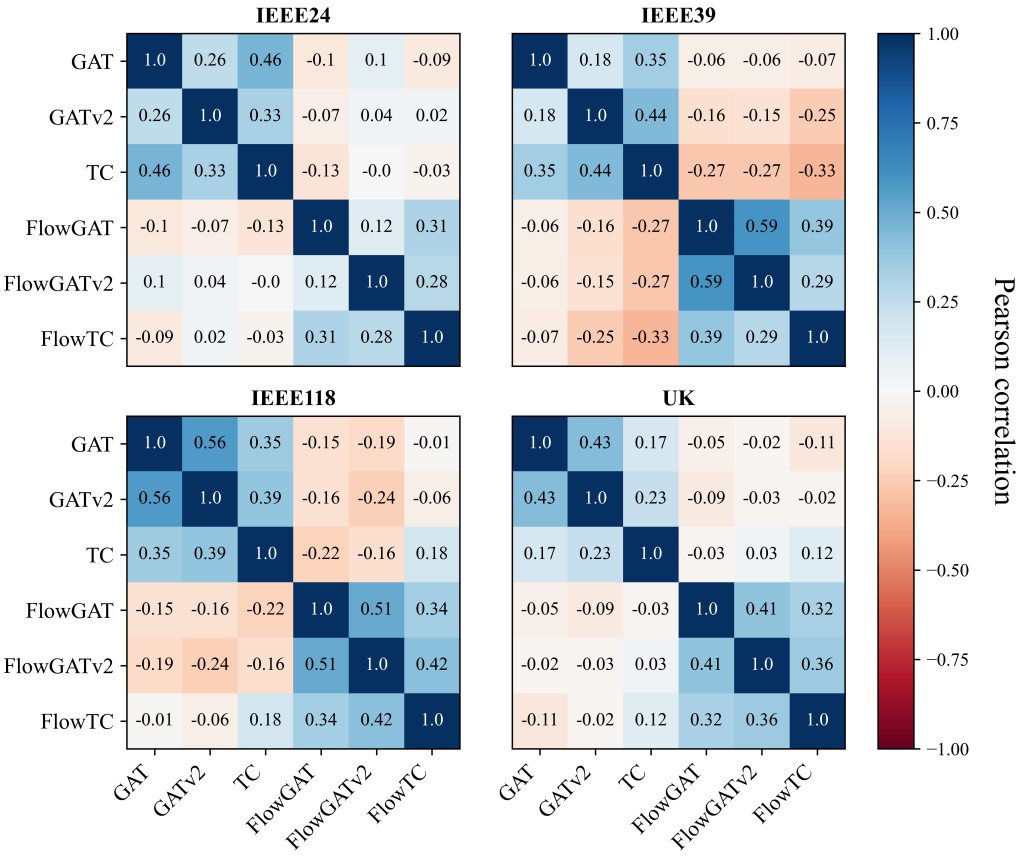

Figure 6: The Pearson correlation between the (flow) attention weights of different models on the four test systems from the PowerGraph dataset, which were also used in the experiments in Sec. 6.

Table 5: **Test set balanced accuracy** (%,↑) and **F1-score (macro-averaged,** %,↑) of state-of-the-art Graph Transformer models in standard configuration for the cascading failure analysis (multiclass classification) on four different power grid test systems from the PowerGraph dataset. Reported results represent the average over five training runs with different random seeds along with the standard deviation. The best result for each test system is marked in bold.

| | IEEE24 | | IEEE39 | | IEEE118 | | UK | |
|---|---|---|---|---|---|---|---|---|
| MODEL | Bal. Acc. (↑) | Macro-F1 (↑) | Bal. Acc. (↑) | Macro-F1 (↑) | Bal. Acc. (↑) | Macro-F1 (↑) | Bal. Acc. (↑) | Macro-F1 (↑) |
| SAT | 99.6 ± 0.3 | 99.0 ± 0.6 | 99.4 ± 0.2 | 98.8 ± 0.5 | 99.8 ± 0.1 | 99.6 ± 0.1 | 99.4 ± 0.1 | 98.9 ± 0.3 |
| GPS | 99.2 ± 0.2 | 98.5 ± 0.4 | 98.6 ± 0.4 | 97.4 ± 0.7 | 99.8 ± 0.1 | 99.5 ± 0.2 | 99.2 ± 0.1 | 98.6 ± 0.3 |
| Exphormer | 99.3 ± 0.3 | 98.8 ± 0.5 | 98.1 ± 0.8 | 97.2 ± 1.5 | 99.0 ± 0.6 | 98.9 ± 0.7 | 98.7 ± 0.4 | 98.3 ± 0.3 |

Table 6: Test set results for flow-attentional GNNs and their standard-attentional base models on popular standard graph datasets beyond the flow graph context. The performances are averaged over 10 runs with different random seeds. The best results for each dataset are marked in bold.

| | Cora | CiteSeer | PubMed | ogbg-molhiv |
|---|---|---|---|---|
| **MODEL** | Acc. (%,↑) | Acc. (%,↑) | Acc. (%,↑) | ROC-AUC (%,↑) |
| GAT | **81.2 ± 1.1** | 69.1 ± 0.8 | 76.6 ± 0.5 | 74.9 ± 3.7 |
| **FlowGAT** | 80.3 ± 0.8 | 67.7 ± 1.3 | **77.4 ± 0.9** | 75.9 ± 0.6 |
| GATv2 | 79.6 ± 1.4 | **69.6 ± 0.7** | **77.4 ± 0.4** | 74.5 ± 1.9 |
| **FlowGATv2** | 80.4 ± 1.0 | 67.7 ± 1.9 | 76.7 ± 1.1 | 75.8 ± 1.5 |
| TC | 78.2 ± 1.3 | 67.3 ± 1.2 | 75.9 ± 0.7 | 77.5 ± 0.7 |
| **FlowTC** | 77.3 ± 0.6 | 67.3 ± 1.1 | 74.6 ± 0.5 | **77.7 ± 0.7** |

in normalization causes the flow-attentional GNNs to learn attention patterns different to those of their corresponding base models, resulting in performance improvements as demonstrated by our experiments.

### G.2 Transformers on PowerGraph

Graph Transformers have recently emerged as a promising alternative to message-passing GNNs (Shehzad et al., 2024). These models include global attention modules in addition to local message-passing layers. This enables a better modeling of long-range interactions and reduces over-smoothing. While showing better empirical performance across many graph datasets and learning tasks, the quadratic complexity of global attention mechanisms limits their scalability to larger graphs (Behrouz & Hashemi, 2024), e.g., real-world power grids with thousands of nodes, particularly for real-time decision making.

In Tab. 5, we report additional experimental results on the PowerGraph dataset (cascading failure analysis) with state-of-the-art Transformer models for graph learning: SAT (Chen et al., 2022), GraphGPS (Rampášek et al., 2022), and Exphormer (Shirzad et al., 2023). Thereby, we train all three models with 3 layers and a hidden dimension of 32, and adopt the remaining hyperparameters from the original publications. On all test systems, these models perform considerably better compared to all message-passing GNNs listed in Tab. 1. However, the superior performance comes at the cost of increased computational complexity. Depending on the task at hand, it is therefore still very valuable to achieve high performance with pure local message-passing GNNs, such as the flow-attentional GNNs.

Additionally, note that many Graph Transformers rely on message-passing GNNs to learn local node representations. Therefore, they could also be used with a flow-attentional GNN, e.g., FlowGAT, FlowGATv2, or FlowTC. In future work, we want to investigate the usage of flow-attentional GNNs compared to other local message-passing GNNs within Transformer frameworks like GraphGPS, since we believe that their performance on flow graph datasets could be further improved in this way.

### G.3  Experiments on Standard Graph Datasets

In Sec. 5.4, we argued that flow-attention might be less effective on graph datasets outside of the flow graph context, because these graphs do not require an information distribution across outgoing neighbors. In addition to that, FlowGNNs might map non-isomorphic but equivalent informational graphs to different representations due to Theorem 5.4. On the other hand, flow-attention could be beneficial on graphs with repetitive node features (e.g., molecular graphs) due to Lemma 5.2.

To complement this discussion, we performed additional experiments with flow-attentional GNNs on standard graph datasets, including node classification on the widely used citation networks Cora (McCallum et al., 2000), CiteSeer (Sen et al., 2008), and PubMed (Namata et al., 2012), as well as graph classification on the molecular property prediction dataset ogbg-molhiv (Hu et al., 2020). Thereby we compared FlowGAT, FlowGATv2, and FlowTC to the corresponding standard-attentional baselines.

For all datasets, we utilized public data splits and optimized hyperparameters on the validation set using the TPE algorithm from Optuna (Akiba et al., 2019). For the ogbg-molhiv, we used sum pooling followed by a multi-layer perceptron for graph-level readout. We report the test set results for the best hyperparameter configurations of each model in Tab. G.3.

On the citation networks, the performance of flow-attentional GNNs is slightly lower than that of their standard-attentional baseline in most cases. In contrast, they perform slightly better compared to their standard-attentional counterparts on ogbg-molhiv. These observations are consistent with our discussion on the applicability of FlowGNNs beyond flow graphs in Sec. 5.4.

### G.4  Efficiency Comparison

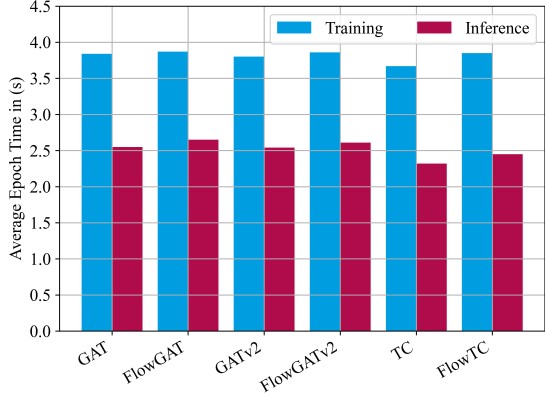

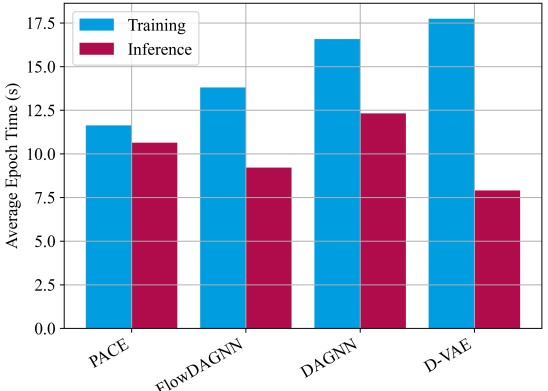

Figure 7:  The training and inference times for processing the training set of the IEEE24 dataset from the PowerGraph benchmark, averaged over 10 runs.

Figure 8:  The training and inference times for processing the training set of the CktBench-101 dataset from the Open Circuit Benchmark, averaged over 10 runs.

**Undirected Graphs.** We compare the average training and inference times of the considered models for processing the training set of the IEEE24 dataset from the PowerGraph benchmark using a batch size of 64 (see Fig. 7). We do not observe significant differences in computational efficiency between any flow-attentional GNN and its standard-attentional counterpart.

**DAGs.** We compare the average training and inference times of the considered directed acyclic GNN models for processing the training set of the CktBench-101 dataset using a batch size of 64 (see Fig. 8). Thereby, we compare our implementation of FlowDAGNN against the original implementations from the authors of the baseline models. While PACE is the most efficient model, FlowDAGNN only shows a slightly higher training time. DAGNN and D-VAE (using bidirectional processing, see App. B) appear to be slightly less

efficient than FlowDAGNN. Note that these differences could be caused not only by architectural but also by implementation differences.

All efficiency experiments were carried out on NVIDA V100 GPUs.

