# OpenReview forum: "Flow-Attentional Graph Neural Networks"
_TMLR — Accepted by TMLR_

### Review · Reviewer_4Amk · 2025-07-15

**Summary Of Contributions:**

Existing GNNs do not consider the conservation law inherent in graphs associated with a flow of physical resources, such as electrical current in power grids or traffic in transportation networks. To address this, this work proposes flow attention, which adapts existing graph attention mechanisms to satisfy Kirchhoff’s first law. The authors also discuss how this modification influences the expressivity and identify sets of non-isomorphic graphs that can be discriminated by flow attention but not by standard attention.

**Audience:**

Yes

**Broader Impact Concerns:**

I have no concerns.

**Claims And Evidence:**

Yes

**Requested Changes:**

Please refer to the weaknesses above and revise the paper accordingly.

**Strengths And Weaknesses:**

Strengths
1. The motivation of this research is intriguing, as the flow mechanism is important to understand the nature of many real graphs.
2. The paper is generally written well. It is easy to understand as it is self-contained and equipped with good visualizations such as Figure 1.
3. The proposed method shows good empirical performance on various datasets.

Weaknesses
1. Most GNNs are designed for undirected graphs. If we naturally extend them to directed graphs, I think Eq. (2) should take the information of outgoing neighbors as well as incoming neighbors, possibly with a simple way to integrate the two types of messages.
2. Regarding W1, you may consider GNNs designed specifically directed graphs to strengthen your claim:
    1. https://arxiv.org/abs/2004.13970
    2. https://proceedings.neurips.cc/paper/2020/hash/cffb6e2288a630c2a787a64ccc67097c-Abstract.html
    3. https://papers.neurips.cc/paper_files/paper/2021/file/e32084632d369461572832e6582aac36-Paper.pdf
3. Regarding W1, if we change the definition of a GNN to taking the information of outgoing neighbors as well, is Lemma 4.1 satisfied?
4. The experiments in Section 6.1 seem irrelevant to the main claim of the paper, as the datasets are undirected. There are few mentions on undirected graphs in the method section.
5. Graph transformers are being widely studied, and there are many recent models that are designed to address the limitations of early versions, e.g., dense attention, computational complexity, etc. I believe these models should be included in this work.

---

> ### Author Response · Authors · 2025-07-23
> **Response to Reviewer 4Amk (1/2)**
>
> Dear Reviewer,
>
> Thank you very much for your insightful comments and suggestions. First, since weaknesses (1) - (4) are related to GNNs for directed graphs, we would like to make clear that we did not design the flow-attention mechanism to adapt attention-based GNNs to directed graphs. In fact, both attention-based GNNs and their flow-attentional variants can be applied to general graphs, including undirected and directed graphs, because they are spatial-based GNNs and do not require the adjacency matrix to be symmetric.
>
> Instead, we designed the flow-attention mechanism to improve the performance of attention-based GNNs on flow graphs, which are defined as graphs associated with a flow that satisfies Kirchhoff’s first law (see Eq. 1). In these graphs, each flow value is indeed associated with a directed edge. However, this does not mean that the graph itself cannot be undirected, which simply means that for every directed edge, there is another directed edge in the opposite direction. For example, a power grid is typically described by an undirected graph because the electrical current can, in principle, flow in both directions between connected nodes (buses). The net direction of the active power flow ultimately depends on the operating conditions.
>
> Point-to-point answers to weaknesses (1)-(5):
>
> (1) As you point out correctly, most GNNs are primarily designed for undirected graphs. However, most spatial-based GNNs (e.g., GIN, GAT, GATv2, GraphTransformer) can also naturally handle directed graphs with asymmetric adjacency matrices. Thereby, the fact that they only take into account incoming neighbors might indeed be a drawback. However, although the flow-attentional GNNs proposed in our paper only aggregate messages from incoming neighbors, they take into account the number of outgoing neighbors when calculating the flow-attention weights (since the attention scores are normalized across outgoing neighbors). In this way, we are already including information about the outgoing neighbors into the message-passing.
>
> (2) Thank you for these interesting suggestions. If we understand correctly, the models described in these papers extend spectral-based GNNs to directed graphs, either by using first- and second-order proximities (DGCN/DIGCN) or by encoding the asymmetry of directed graphs via a complex-valued Hermitian matrix (MagNet).
> Although these models are interesting, they are not attention-based, which is why we cannot integrate the flow-attention mechanism into them. Furthermore, we do not believe that including these models as additional baselines in our study would add significant value for the reader. The reason for this is that these models are not particularly suitable for the considered datasets. We treat power grids as undirected graphs with symmetric adjacency matrices, which means that there is no need for GNNs to account for asymmetry in the graph. Furthermore, the CktBench101 dataset contains directed acyclic graphs (DAGs), a special case of directed graphs, which can be modeled better by GNNs specifically tailored to DAGs. These special DAG models are already included in our study (DVAE, DAGNN, PACE, DAGformer).
>
> (3) Lemma 4.1 holds for general multisets, regardless of whether the contained nodes are incoming or outgoing neighbors. What makes two node multisets with the same distribution indistinguishable for attention-based GNNs is that they perform a weighted mean aggregation by normalizing attention scores across incoming neighbors. In contrast, flow-attentional GNNs perform a weighted sum aggregation (by normalizing attention scores across outgoing nodes), which makes two node multisets with the same distribution distinguishable (see Lemma 5.2).
>
> (4) Attention-based GNNs, such as GAT, GATv2, or GT, can naturally handle directed as well as undirected graphs, and this also applies to the proposed flow-attentional variants. In Sec. 4.1 and 5.1, we are therefore defining attention- and flow-attention-based GNNs on general graphs, including directed and undirected graphs. As stated at the beginning, power grids are usually represented by undirected graphs, since the electrical current can flow in both directions between connected nodes (buses), depending on the operating conditions. Our approach aims to better capture this bidirectional power flow via the flow-attention weights, which are designed to satisfy Kirchhoff’s first law. Therefore, we believe that the experiments in Section 6.1 are highly relevant, especially for researchers in the field of ML-based power grid operation.

---

> ### Author Response · Authors · 2025-07-23
> **Response to Reviewer 4Amk (2/2)**
>
> (5) Thank you for this suggestion. Please note that the flow-attention mechanism is a local attention mechanism for message-passing GNNs. Maybe we caused some confusion by using the Graph Transformer (GT) model in our study [1]. However, the GT model is in fact a message-passing GNN that works similarly to GAT, but uses a different way to compute local attention scores, namely using the attention mechanism from the vanilla Transformer [2] (see App. C). Therefore, GT works differently from recent Transformer-based models, such as Structure-aware Transformer (SAT) [3], GraphGPS [4], and Exphormer [5]. These models are more complex as they utilize global attention and include positional/structural encodings. Since flow-attention is a local attention mechanism, we cannot easily integrate it into Transformer-based models. In our power grid experiment, we therefore focused on comparing with popular message-passing GNNs, which are often part of Transformer-based models that combine local message-passing with global attention, e.g., GraphGPS. However, we understand that comparing our approach to Transformer-based models could be valuable for the reader, and we are currently working on including the models from Ref. [3-5] as additional baselines in the revised version of our manuscript. We would appreciate it if you could briefly tell us if these are the models you had in mind.
>
> With kind regards,
>
> The Authors
>
> [1] Shi, Yunsheng, et al. "Masked label prediction: Unified message passing model for semi-supervised classification." arXiv preprint arXiv:2009.03509 (2020).
>
> [2] Vaswani, Ashish, et al. "Attention is all you need." Advances in neural information processing systems 30 (2017).
>
> [3] Chen, Dexiong, Leslie O’Bray, and Karsten Borgwardt. "Structure-aware transformer for graph representation learning." International conference on machine learning. PMLR, 2022.
>
> [4] Rampášek, Ladislav, et al. "Recipe for a general, powerful, scalable graph transformer." Advances in Neural Information Processing Systems 35 (2022): 14501-14515.
>
> [5] Shirzad, Hamed, et al. "Exphormer: Sparse transformers for graphs." International Conference on Machine Learning. PMLR, 2023.

---

> > ### Comment · Reviewer_4Amk · 2025-07-30
> >
> > Thank you for the clarification. I had initially assumed that Section 4 was dedicated to directed graphs, including Lemma 4.1, and that the motivation for this work came from the use of GNNs on directed graphs. I now understand that the purpose of Section 4 is to introduce attention-based GNNs as a precursor to the main discussion on flow graphs.
> >
> > I have no further concerns. I hope the authors will consider further improving the clarity of the paper regarding the scope and the use of graph transformers in the final version.

---

### Review · Reviewer_VkJn · 2025-07-16

**Summary Of Contributions:**

This paper presents an attention mechanism for graph learning tasks, which is inspired by Kirchhoff's first law. The main difference between this attention mechanism and previous mechanisms is that it normalizes attention scores across outgoing neighbors instead of incoming ones. The authors show that there exists scenarios where the proposed mechanism is more expressive than existing mechanisms (e.g., neighborhoods with the same distribution). The mechanism is also extended to directed acyclic graphs (DAGs) and the FlowDAGNN model is proposed. The proposed mechanism is evaluated on power grids and electronic circuit datasets that contain undirected graphs and DAGs, where it improves the performance of the base GNNs.

**Audience:**

Yes

**Broader Impact Concerns:**

There are no concerns.

**Claims And Evidence:**

Yes

**Requested Changes:**

The requested changes listed below are related to the aforementioned weaknesses, and they are essential to meet the standards required for acceptance:

- Compare the learned attention scores against those learned by existing models such as GAT or GATv2. Train the model on some toy dataset and visualize the learned attention coefficients to validate that nodes can actually attend to their most important neighbors.

- Discuss whether GNN models, such as GAT, equipped with the proposed attention mechanism can be applied to standard graph learning datasets, and conduct experiments on a few datasets (e.g., node classification or graph classification).

- Compare the proposed method against more recent baselines that reflect the current state-of-the-art.

- Provide some further explanation about the proposed attention mechanism and why the authors chose to normalize scores across outgoing edges.

**Strengths And Weaknesses:**

Strengths:

- The proposed mechanism is theoretically well grounded. It is shown that the proposed attention mechanism can distinguish neighborhoods with the same distribution (which cannot be distinguished by existing mechanisms). It is also shown that the FlowDAGNN model can distinguish non-isomorphic DAGs which cannot be distinguished by existing GNNs for DAGs.

- The empirical results demonstrate that the proposed mechanism is effective. The proposed approach improves the performance of the base GNN models on the PowerGraph datasets. Furthermore, FlowDAGNN outperforms all the baselines on the Ckt-Bench101 dataset.

- The paper is well-written and easy to read. The introduction section builds the necessary motivation and context for the work.

Weaknesses:

- Some qualitative results are missing from the paper. The authors could extract and visualize the learned attention coefficients on some toy dataset to validate the proposed method. It is also not clear whether the learned attention weights are correlated with those of the base models such as GAT or GATv2.

- It is not clear whether GNN models equipped with the proposed attention mechanism can be applied to standard node and graph classification or regression datasets (e.g., citation networks, molecules, etc), and whether those models would be effective or not. Some empirical results are missing from the paper.

- The current baseline methods (in Table 1) appear to be outdated. Including more recent and competitive baselines would strengthen the paper and is also important to validate the significance of the results.

- Even though it is clearly mentioned that the mechanism is inspired by Kirchhoff's first law, some further explanation is needed for why attention scores are normalized across outgoing edges. Couldn't these normalized coefficients be learned by some model that doesn't normalize the coefficient at all?

---

> ### Author Response · Authors · 2025-07-24
> **Response to Reviewer VkJn (1/2)**
>
> Dear Reviewer,
>
> Thank you very much for your insightful comments and suggestions. Here are our point-to-point answers:
>
> (1) Fig. 3 in Appendix D of our manuscript shows a toy example that visualizes the difference between standard attention weights and flow-attention weights utilizing the example DAGs from Fig. 1. While the (standard) attention weights are always the same for both graphs, the flow-attention weights are different due to the different normalization. We would appreciate it if you could let us know whether this example meets your expectations. \
> Additionally, we calculated the correlation between the (flow-)attention weights of the different models on the PowerGraph datasets. We observed small to medium positive correlations between GNNs using the same type of attention mechanism (e.g., GAT <-> GATv2, or FlowGAT <-> FlowGATv2) on all datasets. Between GNNs using different types of attention mechanisms (e.g., GAT <-> FlowGAT, or GT <-> FlowGT), we observed no significant correlations on IEEE24 and UK, but small negative correlations on IEEE39 and IEEE118. These small negative correlations could be explained by a higher number of relations between nodes with a strong degree difference, leading to small standard attention weights vs. large flow attention weights or vice versa, depending on the edge direction. We will include these additional results in the revised version of our manuscript.
>
> (2) The proposed attention mechanism can technically be applied to all kinds of datasets  (they do not need to be associated with a physical flow). However, we designed the approach specifically for flow graphs, which means that it might be less effective on other graph datasets. As pointed out in the introduction of our manuscript, some non-isomorphic graph structures might be different in the context of flow graphs (e.g., electronic circuits) but equivalent in the context of informational graphs (e.g., citation graphs). In such informational graph datasets, it would be advantageous to map these equivalent (but non-isomorphic) graphs to the same representation, which would not be ensured by a flow-attentional GNN. \
> On the other hand, the flow-attention mechanism has the advantage over standard attention that it can distinguish between node multisets with the same distribution (Lemma 5.2). This could possibly lead to performance improvements on graph datasets where certain node features occur repeatedly, which is often the case in flow graph applications (e. g., multiple identical components in electronic circuits), but can also be the case outside of the flow graph context (e. g., multiple identical atoms in molecules). \
> To sum up, we think that performing experiments on standard graph datasets, such as citation graphs or molecules, is out of the scope of our paper because we specifically introduced the flow attention mechanism in the context of flow graphs, which is already an important research domain in itself. However, we want to add the above discussion in the revised version of our manuscript, since we agree that it could be of great value for readers who are not involved with the application areas covered by our paper.

---

> > ### Author Response · Authors · 2025-07-24
> > **Response to Reviewer VkJn (2/2)**
> >
> > (3) Thank you for this suggestion. Most of the recent GNN models are Transformer-based, e.g., Structure-aware Transformer (SAT) [1], GraphGPS [2], or Exphormer [3]. These models are more complex than message-passing GNNs as they utilize global attention and include positional/structural encodings. \
> > Please note that the flow-attention mechanism is a local attention mechanism for message-passing GNNs, which are often part of larger Transformer-based models that combine local message-passing with global attention, e.g., GraphGPS. In our power grid experiments, we therefore focused on comparing flow-attentional GNNs to their corresponding standard-attentional base models (GAT, GATv2, GT) and other popular message-passing GNNs (GCN, GraphSAGE, GIN). Maybe we caused some confusion by using the Graph Transformer (GT) model in our study [4]. The GT model is in fact a message-passing GNN that works similarly to GAT, but uses a different way to compute local attention scores, namely using the attention mechanism from the vanilla Transformer [5] (see App. C). Therefore, GT works differently from recent Transformer-based models. \
> > Nevertheless, we also understand that a comparison to recent Transformer-based models could be useful to further validate our approach, and we are currently working on including additional baselines in the revised version of our manuscript (see also point (5) in our answer to Reviewer 4Amk).
> >
> > [1] Chen, Dexiong, Leslie O’Bray, and Karsten Borgwardt. "Structure-aware transformer for graph representation learning." International conference on machine learning. PMLR, 2022. \
> > [2] Rampášek, Ladislav, et al. "Recipe for a general, powerful, scalable graph transformer." Advances in Neural Information Processing Systems 35 (2022): 14501-14515. \
> > [3] Shirzad, Hamed, et al. "Exphormer: Sparse transformers for graphs." International Conference on Machine Learning. PMLR, 2023. \
> > [4] Shi, Yunsheng, et al. "Masked label prediction: Unified message passing model for semi-supervised classification." arXiv preprint arXiv:2009.03509 (2020). \
> > [5] Vaswani, Ashish, et al. "Attention is all you need." Advances in neural information processing systems 30 (2017).
> >
> > (4) The normalization of attention scores across outgoing neighbors ensures that a node’s message cannot be arbitrarily duplicated, which aligns with the concept of resource conservation described by Kirchhoff’s first law. In other words, with the flow-attention mechanism, a model learns how to distribute a node’s message among its outgoing neighbors instead of learning which incoming messages it has to attend to (which is the case for normalizing across incoming messages in standard attention). \
> > Moreover, the normalization across outgoing neighbors has the positive side-effect that it makes node multisets with the same distribution distinguishable, because the node aggregation corresponds to a weighted sum instead of a weighted mean (see Lemma 5.2). \
> > Finally, we demonstrated that the standard attention mechanism with normalization across incoming messages leads to the problem of indistinguishable DAGs (see Fig. 1 and 3). Flow-attention solves this problem as it allows us to distinguish between any DAG and its computation tree (Theorem 5.4). \
> > Regarding your question, we believe that the normalization of the attention scores in general is also very important for numerical stability during training.
> >
> > With kind regards, \
> > The Authors

---

### Review · Reviewer_S68T · 2025-07-21

**Summary Of Contributions:**

This paper presents flow attention, a new attention mechanism for graph neural networks. Flow attention is designed to capture the conservation laws in graphs so that the learned representations can be implanted with physical resource flows such as electric circuits or traffic networks. Theoretical analysis is provided to show flow attention can identify cases that standard attention cannot. Experimental results show the proposed flow attention performs well on electronic circuit and power grid datasets.

**Audience:**

Yes

**Broader Impact Concerns:**

NIL.

**Claims And Evidence:**

Yes

**Requested Changes:**

The authors are suggested to address the points listed under "Weaknesses."

**Strengths And Weaknesses:**

Strengths:
1. The problem tackled in this paper is essential in graph neural networks and corresponding applications.
2. The theoretical analysis provided in the paper shows the effectiveness of the paper to some extent.

Weaknesses:
1. The proposed approach is compared with some standard attention-based GNNs, including GAT, GATv2, and Graph Transformer. The authors are suggested to compare the proposed approach to other standard message-passing GNNs, e.g., those released after 2023.
2. GNNs built upon partial/ordinal differential equations (P/ODE) can also capture the flow properties in the graph. How does the proposed flow attention perform when compared with these P/ODE GNNs?
3. Some case studies should have been presented in the paper to clearly show the properties of the proposed flow attention. It would be great if the authors could show examples from real-world test datasets that can demonstrate how the proposed flow attention is different from standard attention mechanisms and other message-passing paradigms when learning representations. The reasons why the proposed flow attention is better than existing approaches can also be demonstrated with examples.
4. Theoretical analysis shows that the proposed flow attention is more expressive than standard graph attention mechanisms in DAG cases, which limits the contributions of the paper.
5. The overall design of flow attentions is simple despite being effective (row normalization in standard attentions versus column normalizations in flow attentions). The authors are suggested to find other ways to introduce this new attention mechanism and manifest its significance.
6. The generalizability/learning capabilities of the proposed flow attention mechanism are not tested with other types of graph data.

---

> ### Author Response · Authors · 2025-07-28
> **Response to Reviewer S68T (1/2)**
>
> Dear Reviewer,
>
> Thank you very much for your insightful comments and suggestions. Here are our point-to-point answers:
>
> (1) Thank you for this suggestion. Most of the recent GNN models are Transformer-based, e.g., Structure-aware Transformer (SAT) [1], GraphGPS [2], or Exphormer [3]. These models are more complex than message-passing GNNs as they utilize global attention and include positional/structural encodings. \
> Please note that the flow-attention mechanism is a local attention mechanism for message-passing GNNs, which are often part of larger Transformer-based models that combine local message-passing with global attention, e.g., GraphGPS. In our power grid experiments, we therefore focused on comparing flow-attentional GNNs to their corresponding standard-attentional base models (GAT, GATv2, GT) and other popular message-passing GNNs (GCN, GraphSAGE, GIN). Maybe we caused some confusion by using the Graph Transformer (GT) model in our study [4]. The GT model is in fact a message-passing GNN that works similarly to GAT, but uses a different way to compute local attention scores, namely the attention mechanism from the vanilla Transformer [5] (see App. C). Therefore, GT works differently from recent Transformer-based models. \
> Nevertheless, we also understand that a comparison to recent Transformer-based models could be useful to further validate our approach, and we are currently working on including additional baselines in the revised version of our manuscript (see also point (5) in our answer to Reviewer 4Amk).
>
> [1] Chen, Dexiong, Leslie O’Bray, and Karsten Borgwardt. "Structure-aware transformer for graph representation learning." International conference on machine learning. PMLR, 2022. \
> [2] Rampášek, Ladislav, et al. "Recipe for a general, powerful, scalable graph transformer." Advances in Neural Information Processing Systems 35 (2022): 14501-14515. \
> [3] Shirzad, Hamed, et al. "Exphormer: Sparse transformers for graphs." International Conference on Machine Learning. PMLR, 2023. \
> [4] Shi, Yunsheng, et al. "Masked label prediction: Unified message passing model for semi-supervised classification." arXiv preprint arXiv:2009.03509 (2020). \
> [5] Vaswani, Ashish, et al. "Attention is all you need." Advances in neural information processing systems 30 (2017).
>
> \
> (2) We would greatly appreciate it if you could further specify what you exactly mean with GNNs built upon partial / ordinary differential equations. Are you referring to models like CGNN [6], PDE-GCN [7], and GRAND [8]?
>
> [6] Xhonneux, Louis-Pascal, Meng Qu, and Jian Tang. "Continuous graph neural networks." International conference on machine learning. PMLR, 2020. \
> [7] Eliasof, Moshe, Eldad Haber, and Eran Treister. "Pde-gcn: Novel architectures for graph neural networks motivated by partial differential equations." Advances in neural information processing systems 34 (2021): 3836-3849. \
> [8] Chamberlain, Ben, et al. "Grand: Graph neural diffusion." International conference on machine learning. PMLR, 2021.
>
> \
> (3) Fig. 3 in Appendix D of our manuscript shows a toy example that visualizes the difference between the standard attention mechanism and the flow-attention mechanism utilizing the example DAGs from Fig. 1. While the (standard) attention weights are always the same for both graphs, the flow-attention weights are different due to the different normalization. Therefore, flow-attention can distinguish these two non-isomorphic graphs in contrast to standard attention. In this way, we demonstrated the essential properties of the proposed flow-attention mechanism in a simple example. \
> Building up on this intuition, we empirically showed that flow-attentional GNNs perform better compared to their standard-attentional base models on several real-world datasets (power grids and electronic circuits). To explain these experimental results and further support our claims, we provided a comprehensive theoretical analysis. We showed that flow-attention can distinguish between node multisets with the same distribution (Lemma 5.2), while standard attention cannot distinguish them (Lemma 4.1). Furthermore, we showed that flow-attention can differentiate any DAG from its computation tree (Theorem 5.4), in contrast to standard attention (Corollary 4.2). \
> We would appreciate it if you could further specify your ideas for additional examples or case studies that we could include in our study.

---

> > ### Author Response · Authors · 2025-07-28
> > **Response to Reviewer S68T (2/2)**
> >
> > (4) You are right that the proposed flow attention mechanism is more expressive than standard graph attention in DAG cases. Since you believe that this limits the contribution of our paper, you seem to expect that our model should be more expressive on all kinds of graphs. However, our claim was that flow-attention performs better than standard attention on graphs associated with a physical flow, e.g., power grids and electronic circuits. We did not claim that our model is superior on all kinds of graphs. \
> > Since many flow graphs can be naturally represented as DAGs (e.g., electronic circuits), we believe that the increased expressivity on DAGs is an important property that supports our claim. Furthermore, flow-attention can discriminate between node multisets with the same distribution (Lemma 5.1), which means that it may perform better on tasks where the exact graph structure is more important than statistical and distributional information. This is especially useful when node features are repetitive, which is often the case in flow graph applications (e.g., repeating components in electronic circuits).
> >
> > (5) We thank you for highlighting the effectiveness of our approach despite its simplicity. We would greatly appreciate it if you could let us know what kind of other ways you have in mind to introduce the flow-attention mechanism and further manifest its significance.
> >
> > (6) The proposed attention mechanism can technically be applied to all kinds of datasets  (they do not need to be associated with a physical flow). However, we designed the approach specifically for flow graphs, which means that it might be less effective on other graph datasets. As pointed out in the introduction of our manuscript, some non-isomorphic graph structures might be different in the context of flow graphs (e.g., electronic circuits) but equivalent in the context of informational graphs (e.g., citation graphs). In such informational graph datasets, it would be advantageous to map these equivalent (but non-isomorphic) graphs to the same representation, which would not be ensured by a flow-attentional GNN. \
> > On the other hand, the flow-attention mechanism has the advantage over standard attention that it can distinguish between node multisets with the same distribution (Lemma 5.2). This could lead to performance improvements on graph datasets where certain node features occur repeatedly, which is often the case in flow graph applications (e.g., multiple identical components in electronic circuits), but can also be the case outside of the flow graph context (e.g., multiple identical atoms in molecules). \
> > To sum up, we think that performing experiments on other types of graph datasets, such as citation graphs or molecules, is out of the scope of our paper because we specifically introduced the flow attention mechanism in the context of flow graphs, which is already an important research domain in itself. However, we want to add the above discussion in the revised version of our manuscript, since we agree that it could be of great value for readers who are not involved with the application areas covered by our paper.
> >
> > With kind regards, \
> > The Authors

---

> > ### Comment · Reviewer_S68T · 2025-08-04
> >
> > Dear Authors,
> >
> > Thanks very much for your responses, which have addressed most of my concerns. Here are some further questions that you may clarify.
> > 1. For "GNNs built upon partial/ordinary differential equations," you may compare the proposed method with CGNN, PDE-GCN, GRAND, and other related methods.
> > 2. For additional experiments on other types of graph datasets, I still suggest the authors conduct some simple ones (e.g., those on standard and classical datasets, e.g., Cora, Cite, Pubmed, and others), which may enhance the completeness of this paper.  These results may be listed in the appendix if available.

---

> > > ### Author Response · Authors · 2025-08-15
> > >
> > > Dear Reviewer,
> > >
> > > Thank you for your answer and further suggestions.
> > >
> > > 1) After careful consideration, we have decided not to include these comparisons since they are out of the scope of our paper. Although they are built upon PDEs, the models you mentioned are typically applied to standard graph datasets and node classification tasks, which differ from our focus on flow graphs and graph-level tasks.
> > >
> > > 2) In our second revision, we have added further experimental results on the standard graph datasets Cora, CiteSeer, PubMed, and ogbg-molhiv in App. G.3. These results are consistent with our previous discussion on the applicability of FlowGNNs beyond flow graphs in Sec. 5.4.
> > >
> > > With kind regards,\
> > > The Authors

---

### Author Response · Authors · 2025-08-03
**Revision**

Dear Reviewers,

We would like to express our sincere gratitude for your insightful and constructive feedback on our paper. Based on your valuable comments, we have revised our manuscript and thereby addressed the following points:

**Clarification: Distinction from Graph Transformers**

We improved our wording and emphasized that the proposed flow-attention mechanism is a local attention mechanism for message-passing GNNs. \
Furthermore, we changed the name of the Graph Transformer (GT) model into TransformerConv (TC) (referencing its PyG implementation) to avoid any confusion with the model class of Graph Transformers, which include global attention mechanisms, e.g., GraphGPS [1]. We also added a footnote explaining that TC is actually a message-passing GNN with a local attention mechanism adopted from the vanilla Transformer [2].

**Comparison to State-of-the-Art Graph Transformers: Additional Results and Discussion**

We performed additional experiments on the PowerGraph datasets with the following state-of-the-art Graph Transformer models: GraphGPS [1], SAT [3],  and Exphormer [4]. In addition to local message-passing layers, these models also include global attention modules. As expected, our experiments show that they perform better than the pure message-passing GNNs we compared in our power grid experiments. \
However, since many recent Graph Transformers rely on message-passing GNNs for learning local node representations, it is also possible to use them with flow-attentional GNNs, e.g., FlowTC. Investigating the influence of using flow-attentional GNNs compared to other message-passing GNNs within Graph Transformer frameworks like GraphGPS is an interesting direction for future work. \
We added Section G.2 in the Appendix, including these new results and a corresponding discussion.

**Correlation between Standard and Flow Attention Weights**

We added Section G.1 in the Appendix with the new Figure 6 showing the correlations between standard and flow attention weights on the four test systems contained in the PowerGraph dataset, which helps to better understand the relationship between them.

**Applicability Beyond Flow Graphs**

We added Subsection 5.4 at the end of Section 5 highlighting the universal applicability of our flow-attention mechanism despite its motivation related to the flow graph domain, and discussing its potential effectiveness on standard graph datasets (e.g., citation graphs or molecules).

UPDATE: At the suggestion of Reviewer S68T, we have added further experimental results on the widely used standard graph datasets Cora, CiteSeer, PubMed, and ogbg-molhiv in App. G.3. These results are consistent with our previous discussion on the applicability of FlowGNNs beyond flow graphs in Sec. 5.4.

**Renaming of Section 4**

We changed the name of Section 4 into “Graph Attention and its Limitations” to clarify its purpose to introduce standard graph attention as a precursor for the following discussion on flow graphs.

We would like to thank you once again for your valuable time and effort in reviewing our manuscript. We believe your comments have significantly contributed to the improvement of our work, and we greatly appreciate your support.

With kind regards, \
The Authors

[1] Rampášek, Ladislav, et al. "Recipe for a general, powerful, scalable graph transformer." Advances in Neural Information Processing Systems 35 (2022): 14501-14515. \
[2] Vaswani, Ashish, et al. "Attention is all you need." Advances in neural information processing systems 30 (2017). \
[3] Chen, Dexiong, Leslie O’Bray, and Karsten Borgwardt. "Structure-aware transformer for graph representation learning." International conference on machine learning. PMLR, 2022. \
[4] Shirzad, Hamed, et al. "Exphormer: Sparse transformers for graphs." International Conference on Machine Learning. PMLR, 2023.

---

### Decision · Action_Editor_ubC7 · 2025-08-30

**Recommendation:** Accept as is

**Audience:**

Yes

**Audience Explanation:**

Both attention and graphs of interests and the findings are relevant.

**Claims And Evidence:**

Yes

**Claims Explanation:**

All reviewers have been convinced after discussions, and I checked and agree.